

# A large-scale image-text dataset benchmark for farmland segmentation

Chao Tao, Dandan Zhong, Weiliang Mu, Zhuofei Du, and Haiyang Wu
School of Geosciences and Info-Physics, Central South University, Changsha 410083, China

*Correspondence to*: Haiyang Wu (245001024@csu.edu.cn)

**Abstract.** Understanding and mastering the spatiotemporal characteristics of farmland is essential for accurate farmland segmentation.  The traditional deep learning paradigm that solely relies on labeled data has limitations in representing the spatial relationships between farmland elements and the surrounding environment. It struggles to effectively model the dynamic temporal evolution and spatial heterogeneity of farmland. Language, as a structured knowledge carrier, can explicitly

express the spatiotemporal characteristics of farmland, such as its shape, distribution, and surrounding environmental information. Therefore, a language-driven learning paradigm can effectively alleviate the challenges posed by the spatiotemporal heterogeneity of farmland. However, in the field of remote sensing imagery of farmland, there is currently no comprehensive benchmark dataset to support this research direction. To fill this gap, we introduced language-based descriptions of farmland and developed FarmSeg-VL dataset—the first fine-grained image-text dataset designed for

spatiotemporal farmland segmentation. Firstly, this article proposed a semi-automatic annotation method that can accurately assign caption to each image, ensuring high data quality and semantic richness while improving the efficiency of dataset construction. Secondly, the FarmSeg-VL exhibits significant spatiotemporal characteristics. In terms of the temporal dimension, it covers all four seasons. In terms of the spatial dimension, it covers eight typical agricultural regions across China, with a total area of approximately 4,300 km². In addition, in terms of captions, FarmSeg-VL covers rich spatiotemporal

characteristics of farmland, including its inherent properties, phenological characteristics, spatial distribution, topographic and geomorphic features, and the distribution of surrounding environments. Finally, we present a performance analysis of vision language models and the deep learning models that rely solely on labels trained on the FarmSeg-VL, demonstrating its potential as a standard benchmark for farmland segmentation. The FarmSeg-VL dataset will be publicly released at https://doi.org/10.5281/zenodo.15099885(Tao et al., 2025).

## 1 Introduction

Farmland has been the foundation of agricultural food security, and accurately monitoring farmland has been crucial for implementing policies such as farmland improvement, enhanced supervision, and planning and control (Sishodia et al., 2020). Currently, the intelligent interpretation of remote sensing images for farmland based on deep learning has become a primary method for farmland monitoring(Li et al., 2023; Tu et al., 2024) .



However, existing farmland remote sensing image segmentation methods mainly follow a label-driven deep learning paradigm, which faces significant bottlenecks in both data and model. Specifically, in terms of datasets, although existing benchmark datasets have contributed to the advancement of farmland segmentation technology to some extent, they rely solely on label-driven deep learning paradigm, which has two main limitations: First, a single label can only drive the model to learn shallow visual features of farmland, which fails to reveal the underlying driving mechanisms affecting the spatial distribution

and temporal evolution of farmland. Additionally, it is difficult to represent the spatial-temporal heterogeneity in complex agricultural environments. Specifically, the surface cover of farmland shows seasonal differences in complete coverage, partial coverage, and no coverage with the growth cycle of crops, while diverse terrain leads to significant geographical differentiation in the spatial distribution of farmland and its associations with surrounding features such as water bodies, buildings, and vegetation. However, existing datasets cannot represent this kind of spatial-temporal heterogeneity, making it difficult for

models to establish the inherent relationships between farmland and its surrounding environment. In terms of model, although technologies such as convolutional neural networks (CNNs), graph convolutional networks (GCNs), and Transformer have significantly enhanced feature representation capabilities, the existing label-driven paradigm inherently has clear theoretical flaws. First, the existing label-driven paradigm to excessively rely on visual cues and neglect the logical connections between farmland and its surrounding environment in complex farmland scenarios. Second, the label struggles to reflect the evolution

of farmland across seasons and growth stages, severely limiting the model's generalization ability in spatiotemporal dynamic scenarios. Therefore, there is an urgent need to break through the theoretical framework of the traditional label-driven deep learning paradigm and explore a new paradigm capable of uncovering the deep semantic logic of farmland.

        With the emergence of vision-language models (VLMs) and their expanding applications across various fields, studies (Devlin et al., 2019; Liu et al., 2023; Wu et al., 2025)have shown that language can reveal deeper semantic clues behind visual

information. This breakthrough makes up for the shortcomings of existing farmland datasets that only rely on label-guided models to handle complex spatiotemporal heterogeneous farmland scenes, making it possible to mine the complex semantic information in farmland remote sensing images and then model the deep inherent logical relationship between farmland and its surroundings. Specifically, language can guide models to capture farmland features across multiple dimensions, including shape and boundaries, phenological characteristics that reflect seasonal changes and crop growth states, spatial layout based

on latitude and longitude, and geographical features such as terrain and landscape morphology. Additionally, language can describe the relative positional relationships between farmland and surrounding features such as water bodies, buildings, and vegetation. By integrating these rich semantic cues, VLMs can better understand and interpret the complexity of farmland.

        However, in remote sensing, many existing image-text datasets struggle to provide detailed captions and precise annotations for specific land features like farmland. As a result, they often fall short of meeting the requirements for high-accuracy farmland

segmentation. For example, the first large-scale remote sensing image-text pair dataset RS5M (Zhang et al., 2024) and the SkyScript dataset (Wang et al., 2024), which contains millions of image-text combinations, although large in scale, provide a relatively rough description of farmland and fail to deeply describe the specific characteristics of the farmland. In addition, although the manually annotated dataset RSICap (Hu et al., 2023) provides scene-level semantic descriptions, it lacks a refined



depiction of the characteristics of the farmland itself, making it difficult to meet the model's need for deep semantic information
extraction of the farmland. In contrast to the methods mentioned above, ChatEarthNet (Yuan et al., 2024) seeks to enhance the
richness of semantic captions for land cover types by employing detailed prompt strategies and leveraging semantic
segmentation labels from ChatGPT and the WorldCover project. However, due to the inherent randomness of automatically
generated captions, these captions tend to emphasize the spatial location of farmland within the image while often lacking
detailed information about its inherent attributes. Although these datasets have contributed significantly to advancing image-
text understanding in remote sensing, most focus on general remote sensing tasks, with only a small portion dedicated to
farmland captions. Moreover, these captions are often neither comprehensive nor in-depth. Existing datasets have not fully
reflected the complexity of farmland and its changing characteristics over time and space. This is particularly evident in high-
precision farmland segmentation tasks, where there is a lack of deep analysis of farmland characteristics and how they behave
in different scenarios.

To address the above issues, this paper constructs the FarmSeg-VL dataset, a dedicated image-text dataset focused on
farmland segmentation, which fully reflects the spatiotemporal characteristics of farmland. FarmSeg-VL covers eight typical
agricultural regions in China and includes data samples from four seasons, filling the gap of spatial and temporal imbalance in
existing datasets. With its extensive geographical coverage and seasonal variations, this dataset ensures effective support for
the learning of various forms of farmland.

The contributions of this paper are as follows:

1) This study constructed the first farmland image-text benchmark dataset, filling the gap in remote sensing image-text
datasets for the farmland-dedicate domain. This dataset includes various types of farmland, and covers a wide spatial and
temporal range, providing a high-value data foundation for the application research of vision language models in the field
of farmland segmentation.

2) We summarize 11 key elements for describing farmland's inherent properties and its surrounding environment, offering
a comprehensive framework for characterizing farmland from multiple perspectives. Additionally, a text template for
describing farmland images was designed, providing an important reference for constructing a language dataset focused
on farmland.

3) This study developed a semi-automated annotation method based on the caption templates constructed in this paper. We
utilize the semi-automated annotation approach to generate mask and rich captions, significantly reducing labor time while
enhancing the authenticity and reliability of the annotations.

4) Extensive experiments have demonstrated that the model trained on the image-text farmland dataset proposed in this paper
significantly improves farmland segmentation performance and exhibits strong transferability, providing a performance
baseline for vision language models in farmland segmentation.





## 2 Review of Existing Remote Sensing Datasets for Farmland Segmentation

### 2.1 Non image-text dataset

Traditional remote sensing dataset for farmland segmentation are mainly annotated with single-label, which can be divided into two categories: dedicated dataset and non-dedicated dataset. The detailed information is provided in Table 1 (where SR refers to Spatial Resolution in meters, and FP refers to Farmland Proportion). Non-dedicated datasets, such as the scene level dataset BigEarthNet (Sumbul et al., 2019), are not very suitable for pixel level farmland segmentation. Piexel-level dataset, such as WorldCover (ESA) (Zanaga et al., 2022), DynamicWorld (DyWorld) (Brown et al., 2022), and LandCover (Karra et al., 2021) , primarily focus on large-scale mapping and macro-level analysis, making them less suitable for fine-grained farmland segmentation. Moreover, Evlab-SS (Wang et al., 2017) focuses on pixel-level classification, but the proportion of farmland pixels is relatively low, and it remains limited in data scale and coverage area. Although GID (Tong et al., 2020), DeepGlobe-LandCover (Demir et al., 2018), and LoveDA (Wang et al., 2022) cover large farmland areas with relatively high pixel proportions, the farmland samples lack diversity. For example, the farmland forms in DeepGlobe-LandCover and LoveDA are mostly regular and contiguous, lacking diversity in farmland representation. While these non-dedicated datasets provide large amounts of data for farmland segmentation, their annotations are relatively coarse. Specifically, in pixel-level farmland segmentation, they struggle to fully cover the complex shapes, distribution patterns, and finer details, such as crop growth stages.

**Table 1 Detailed information on non image-text dataset of farmland.**

| Type | Dataset | Category | SR | Image size | FP | Region |
|---|---|---|---|---|---|---|
| **Non-dedicated datasets** | Evlab-SS | 11 | 0.1-2 | 4500×4500 | 8.77 | / |
| | GID | 15 | 4 | 56×56,112×112, 224×224 | 30.66 | China |
| | DGLC | 7 | 0.5 | 2448×2448 | 57.74 | Thailand, Indonesia, India |
| | LoveDA | 7 | 0.3 | 1024×1024 | 26.79 | Nanjing,Changzhou,Wuhan,China |
| | Bigearthnet | 43 | 10-60 | 120×120 | 12.41 | / |
| **Dedicated datasets** | GFSAD30 | 3 | 30 | / | / | Europe,Middle East,Russia and Asia |
| | VACD | 2 | 0.5 | 512×512 | / | Guangdong,China |
| | WEIMIN | 2 | 0.5-2 | 512×512 | / | Hebei,China |
| | FGFD | 2 | 0.3 | 512×512 | / | Heilongjiang,Hebei,Shanxi,Guizhou,Hubei ,Jiangxi,Xizang,China |

In contrast, dedicated datasets such as GFSAD30 (Phalke and Özdoğan, 2018), WEIMIN (Hou et al., 2023), VACD (Li et al., 2024), and FGFD (Li et al., 2025) are specifically designed for farmland segmentation. These datasets offer high-precision farmland annotationand cover a broader range of farmland forms, crop distributions, and other relevant information. The GFSAD30 dataset has a spatial resolution of 30m, making it suitable for large-scale farmland monitoring, but not for fine-grained farmland segmentation. By contrary,WEIMIN and VACD offer higher resolutions,however, since WEIMIN only covers Hebei and VACD only covers Guangdong in China, the diversity of farmland samples is limited. The FGFD dataset includes farmland samples from multiple geographic regions. However, it does not account for the phenological characteristics of farmland, limiting its ability to capture seasonal variations and crop growth stages. Although these dedicated datasets offer





high annotation accuracy and support fine-grained regional monitoring, their reliance solely on labels to represent farmland's
visual characteristics across different spatiotemporal conditions overlooks its inherent complexity and diversity. As a result,
they struggle to capture the subtle differences and dynamic changes in farmland driven by seasonal variations and
environmental factors.

**2.2 Image-Text Datasets**

Existing remote sensing image-text paired datasets, such as UCM-Captions (Qu et al., 2016), RSICD (Lu et al., 2018), RS5M,
NWPU-Captions (Cheng et al., 2022), RSICap, SkyScript, and ChatEarthNet, have been widely used in remote sensing
research (see Table 2, where CGM denotes Caption Generation Method). However, these datasets are primarily designed for
tasks such as image captioning, scene classification, or image-text retrieval, with limited applicability to farmland segmentation.
This limitation stems from their insufficient in-depth semantic representations of farmland morphological characteristics,
spatial distribution patterns, and contextual relationships with surrounding features. Consequently, these datasets cannot meet
the requirements for fine-grained semantic understanding essential for high-precision farmland segmentation.

Specifically, most of these datasets focus on high-level descriptions of images, such as scene level or object level
characteristics, rather than the detailed semantic annotations needed for fine-grained tasks like farmland segmentation. For
example, in SkyScript, the image caption "land use of farmland" provides only broad classification information without
offering specific details about farmland characteristics, such as shape, boundaries, crop growth stages, or surrounding
environmental features. Similarly, the RS5M dataset provides only brief titles for images, primarily indicating the image source
and land cover categories, without offering detailed descriptions of farmland. Additionally, while some datasets use automated
methods to generate large-scale image-text pairs, these automatically generated datasets often suffer from inconsistent quality.
The generated text frequently lacks detail and contains redundant information, reducing its effectiveness for fine-grained
farmland analysis. For example, in ChatEarthNet, image captions divide each image into four sections—top, bottom, left, and
right—focusing on the proportions of primary and secondary land cover types in each section rather than providing a dedicated
description of farmland. Manually annotated datasets, such as UCM-Captions, RSICD, and NWPU-Captions, provide five
captions per farmland image. However, these descriptions are often repetitive and lack specificity. For example, in UCM-
Captions, farmland is described simply as "There is a piece of farmland," while the remaining four descriptions merely rephrase
this sentence without adding meaningful details. In RSICD, captions are limited to color and location, such as "green" or
"between two forests." NWPU-Captions expands on this slightly by incorporating shape descriptions, like "rectangular," but
still lacks deeper insights into farmland characteristics. Although RSICap includes descriptions related to image quality, its
farmland annotations remain focused on landscape features and surrounding environments, overlooking inherent farmland
attributes. This limited descriptive approach fails to capture farmland's spatiotemporal complexity, making it hard for precise
farmland semantic segmentation.



**Table 2. Detailed information on the image-text dataset.**

| Dataset | Example | CGM | Number | Farmland-related Descriptions |
|---|---|---|---|---|
| UCM-Captions | | manual annotation | 2100 images, with 5 captions per image | 1.There is a piece of farmland.<br>2.There is a piece of farmland.<br>3.It is a piece of farmland.<br>4.It is a piece of farmland.<br>5.Here is a piece of farmland. |
| RSICD | | manual annotation | 10921 images, each with 5 captions | 1.the cream colored and aqua farmland is between two forest.<br>2.the green and white farmland is between two forest.<br>3.the cream colored and aqua farmland is between two forest.<br>4.this farmland with light green parts and bald ones mixes up with those deep green woods.<br>5.some pieces of green farmlands are together. |
| RS5M | | filtering publicly available image-text paired datasets | 5 million images with captions | a satellite image of a farm with a green field |
| NWPU-Captions | | manual annotation | 31500 images, 5 captions per image | 1.Some rectangular farmlands of different colors are neatly lay out on the ground.<br>2.Many neatly arranged dark green, light green and tan mixed rectangular farmlands of different sizes.<br>3.There are some green and uncovered rectangular farmland.<br>4.There are rectangular farmlands of varying sizes.<br>5.There are some green rectangular farmlands distributed neatly. |
| RSICap | | manual annotation | 2585 image-text pair | This is a low-resolution panchromatic satellite image showing a village and farmland. At the bottom of the image, there is a village with dense buildings, and above the village is a large area of farmland, divided into sections by some dirt roads. There is also a body of water in the middle of the farmland. In the image, you can also see an airplane, which was probably captured by the satellite when it was flying over the farmland. |
| SkyScript | | linking remote sensing images with semantics in OSM through geographic coordinates | 2.6 million image-text pairs, each image corresponding to a title describing a single object and a title describing multiple objects | Single-object text: landuse of farmland, crop of cotton<br>Multi-object text: landuse of farmland with crop of cotton |
| ChatEarth Net | | automatically generate GPT through effective prompts | ChatGPT3.5 generates 163488 image-text pairs, ChatGPT-4v generates 10000 image-text pairs containing captions | The image primarily consists of crop fields, which are most dominant across all sections. In the top left, there is a significant expanse of crop fields, with a small area of grass and developed land. Moving to the top right, crop fields continue to dominate, followed by a smaller developed area and grassy patches. In the bottom left, the landscape is mostly covered by crop fields, followed by a few trees and a small amount of grass. The bottom right also exhibits a large area of crop fields, accompanied by a small developed area and a small portion of grass. In the middle section, crop fields are again the main feature, with a small number of trees and a tiny developed area. Overall, the image depicts a landscape predominantly characterized by crop cultivation, with minor presence of developed areas, trees, and grass. |



Although these image-text datasets have achieved certain results in large-scale pre-training tasks, their application in the semantic segmentation of farmland remote sensing images is greatly limited due to the lack of pixel-level annotation for

semantic segmentation and in-depth description of specific tasks such as farmland segmentation. Therefore, to better support farmland segmentation, the dataset needs to be enhanced by including more fine-grained semantic annotations and comprehensively covering the complex features of farmland.

## 3 FarmSeg-VL: A large-scale image-text dataset benchmark for farmland segmentation

### 3.1 Construction of FarmSeg-VL

The construction process of the FarmSeg-VL is shown in Fig. 1, which is mainly divided into three parts: remote sensing image acquisition and processing, caption construction, and semi-automatic annotation. In the part1, we collected high-resolution images (with a resolution of 0.5m-2m) from various typical agricultural regions in China across four seasons to ensure the dataset covers farmland with diverse spatiotemporal features. In the part2, the study synthesized the spatiotemporal characteristics of farmland and summarized 11 key factors related to its inherent properties and the distribution of surrounding

environments. These factors were then used to generate detailed captions, covering aspects such as farmland shape, terrain, sowing situation, and the distribution of surrounding water bodies, vegetation, and buildings. In the part3, a semi-automated manual annotation method was employed to generate corresponding binary masks and a segment of caption for each remote sensing image sample, thus completing the dataset construction.

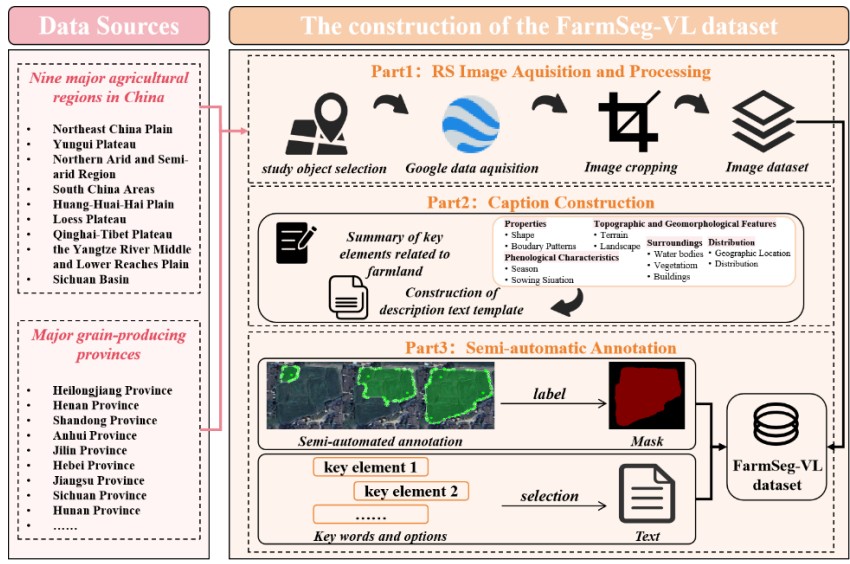

170                                                        **Fig. 1. Dataset construction.**

The FarmSeg-VL dataset, as shown in Fig. 2, consists of three key components: image, mask, and text. Specifically, FarmSeg-VL includes image data from eight major agricultural regions across four seasons, and the image features include



diversity under different imaging conditions. The caption focuses on five attributes of farmland remote sensing images with a total of eleven key features: inherent properties (such as shape and boundary pattern), phenological characteristics (such as season and sowing situation), spatial distribution (such as distribution and geographic location information), topographic and geomorphic features (such as terrain and landscape), and distribution of surrounding environments (such as buildings, water bodies, and vegetation).

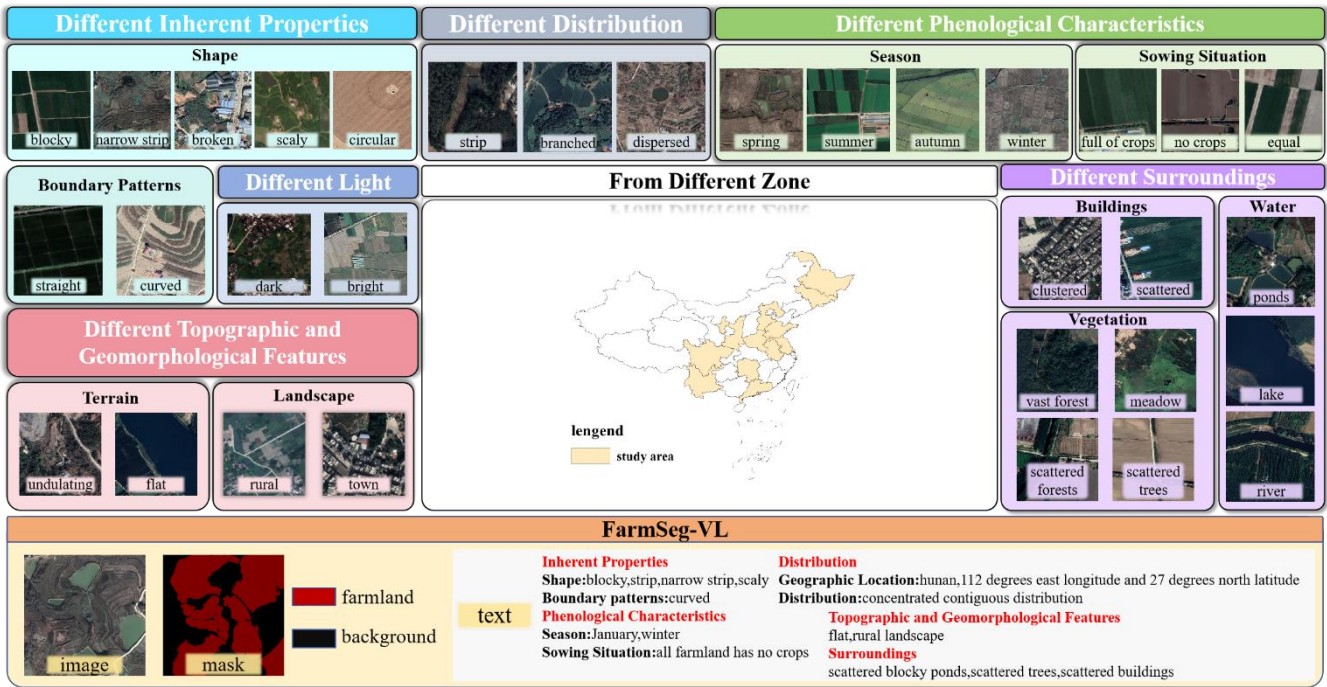

Fig. 2. Attribute annotation and spatiotemporal distribution of FarmSeg-VL.

**1) RS Image Acquisition and Processing**

Farmland exhibits significant spatiotemporal dynamics and fragmented distribution characteristics, and presents diverse spatial patterns due to regional differences. For example, the land in the Northeast China Plain is flat and fertile, and the farmland has the characteristics of concentrated distribution and regular shape, while the Yungui Plateau in China has complex terrain and diverse climate, and the farmland has the characteristics of dispersed distribution and fragmented shape. The farmland appearance and characteristics of these agricultural areas are unique, which poses different challenges and opportunities for farmland segmentation. This study selected representative agricultural regions based on the spatial distribution and morphological characteristics of farmland. Specifically, based on the spatial aggregation and morphological regularity of farmland, the Northeast China Plain and Huang-Huai-Hai Plain were selected as typical regions characterized by concentrated and regular-shaped farmland. For areas with sloped farmland distribution, the Northern Arid and Semi-Arid Region and the Loess Plateau were chosen as study areas. At the same time, in view of the particularity of farmland morphology, such as narrow and long, striped, and sporadic and fragmented, the South China Areas, Sichuan Basin, Yungui Plateau, and Yangtze





River Middle and Lower Reaches Plain were selected as research areas. The study covers 13 provincial-level administrative regions, including Heilongjiang, Jilin, Ningxia, Hebei, Henan, Shandong, Shaanxi, Anhui, Hunan, Jiangsu, Guangdong, Sichuan, and Yunnan. These regions provide broad spatial coverage, highlight distinct regional characteristics, and are highly

representative and typical of China's diverse agricultural landscapes.

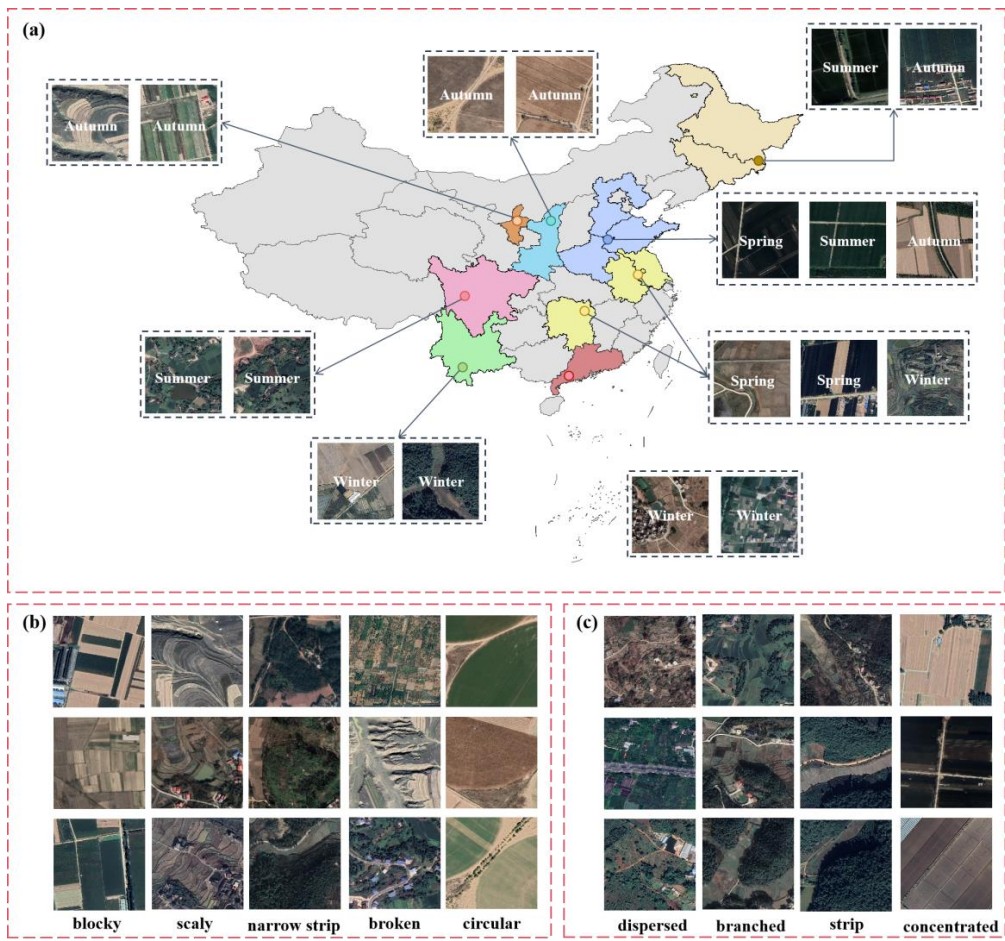

**Fig. 3. Demonstration of the diversity of data samples. (a) Farmland samples from different agricultural regions. (b) Farmland samples with different shapes(c) Farmland samples with varying distribution patterns.**

The data samples diversity are shown in Fig. 3. Specifically, we utilized Bigemap software to acquire high-resolution Google

satellite imagery covering China, including the eight major agricultural regions previously mentioned. The spatial resolution of images ranges from 0.5m to 2m. Additionally, the software enables us to obtain the shooting time of the image. The total coverage area spans approximately 4300 km², ensuring that the dataset covers a broad geographic region and reflects the diverse characteristics of farmland. The images underwent a series of detailed pre-processing steps, including calibration and cropping. During image calibration, we corrected geometric distortions caused by the shooting angle and Earth's curvature,

ensuring spatial consistency across all images. In the cropping process, irrelevant areas were removed, focusing solely on



extracting farmland regions. Additionally, to enhance the dataset's quality, we manually filtered out images affected by cloud or fog cover, stitching artifacts, or overall poor quality, ensuring only high-quality samples remained for analysis. In order to achieve an optimal balance between retaining the detailed features of high-resolution images and improving the efficiency of model training, this study adopted a standardized preprocessing process: all images that passed the quality screening were

uniformly normalized, and a standardized cropping strategy of 512×512 pixels was applied. The size selection was based on the following two considerations. First, to preserve spatial resolution and detail, the 512×512 cropping unit can effectively balance the complete expression of local ground features (such as farmland boundaries and vegetation textures) and the efficient allocation of computing resources. Second, to preserve the integrity of spectral information, the cropped images strictly retain the three visible light bands—red, green, and blue—to ensure the effective transmission of spectral features in

the model. This normalization processing scheme significantly improves the efficiency of batch data processing during model training by unifying the input data dimensions, while avoiding feature learning bias caused by image size differences. After completing these pre-processing steps, a total of 22,605 image samples were selected. These samples span various seasons, regions, cropping statuses, and feature diverse farmland distributions and shapes, ensuring the comprehensiveness and diversity of the dataset. This provides a rich and varied training dataset for the subsequent farmland segmentation.

**2) Caption Construction**

For the caption construction of each farmland sample, this study summarizes key features and keywords for describing farmland from both temporal and spatial perspectives. Temporally, the variations in crop growth stages lead to distinct visual texture differences in farmland across different seasons. Spatially, this study considers the issue at multiple spatial scales. At the macro-regional scale, typical farmland images were collected from various agricultural regions across China. These regions

are not only located in different latitudes and longitudes, but also have different terrains and topography. For instance, in the Northeast China Plain, farmland terrain is flat, while in the South China region, the terrain is predominantly hilly and mountainous, with farmland exhibiting undulating topography. Furthermore, even within the same region, there are differences in landscape features such as the presence of rural areas and towns around the farmland. At the image scale, the spatial distribution of farmland varies significantly depending on its geographical location. For instance, in the Northeast China Plain,

farmland typically follows a concentrated distribution pattern, whereas in South China, farmland tends to be more dispersed. At the same time, the spatial relationship between farmland and other land features is also very complex. For example, water bodies, vegetation, buildings, are all part of the surrounding environment of farmland. Similarly, the shape of the farmland, the boundary shape of the farmland, can all be used as key elements to describe the farmland.

In summary, as shown in Fig. 4, this study categorizes farmland-related attributes into five major aspects: inherent properties,

phenological characteristics, spatial distribution, topographic and geomorphic features and distribution of surrounding environments. The inherent properties include the shape of the farmland and the boundary patterns. Phenological characteristics encompass season and the sowing situation of the farmland. The spatial distribution of farmland not only reflects the geographic location information, but also includes the macro-level distribution of farmland in the image, such as



concentrated contiguous distribution or dispersed distribution. Farmland shape is a very intuitive and important feature in

visual interpretation, closely related to other factors such as terrain, topography, and landscape features, including blocky, striped, or broken. Farmland boundary pattern refers to the spatial shape characteristics of the farmland boundary, primarily manifested in whether its contour lines are relatively straight or exhibit a curved form.

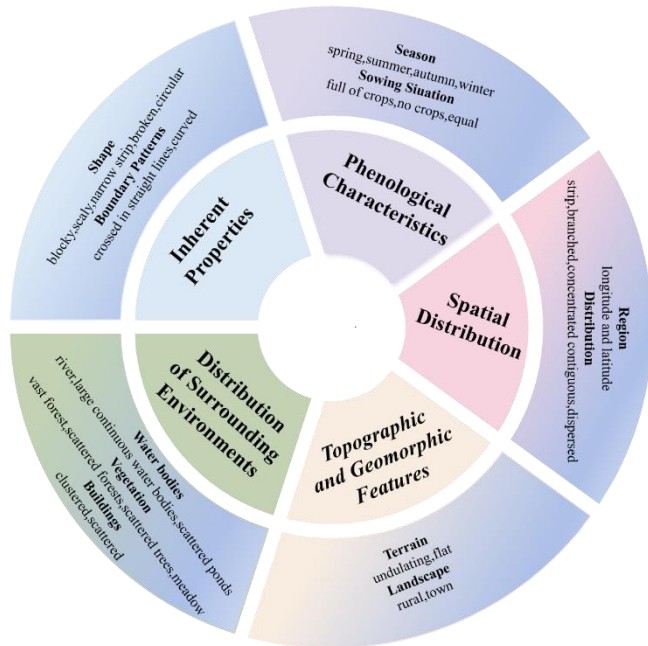

**Fig. 4. Farmland description keywords.**

**3) Semi-Automated Annotation**

Currently, there are two main approaches for constructing remote sensing image-text datasets: one involves automatically generating textual annotations using large language models, while the other relies on manual visual annotation by humans. However, both methods face significant challenges in meeting the high-precision requirements of farmland segmentation. Relying solely on automatic annotations generated by large language models has clear limitations. This approach often

struggles to capture the nuanced and accurate correspondence between images and text. The granularity of captions is often insufficient, resulting in suboptimal accuracy and completeness in the annotation process. While manual annotation can ensure high-quality data, it has significant drawbacks. This approach requires domain experts to invest substantial time and effort, draining valuable resources and leading to extremely low efficiency. To address these challenges, this study proposes and develops a semi-automatic farmland image-text annotation framework. It is important to highlight that this semi-automatic

annotation framework differs from previous methods. In addition to enabling text annotation, it also generates high-quality masks, offering more effective data support for farmland segmentation.



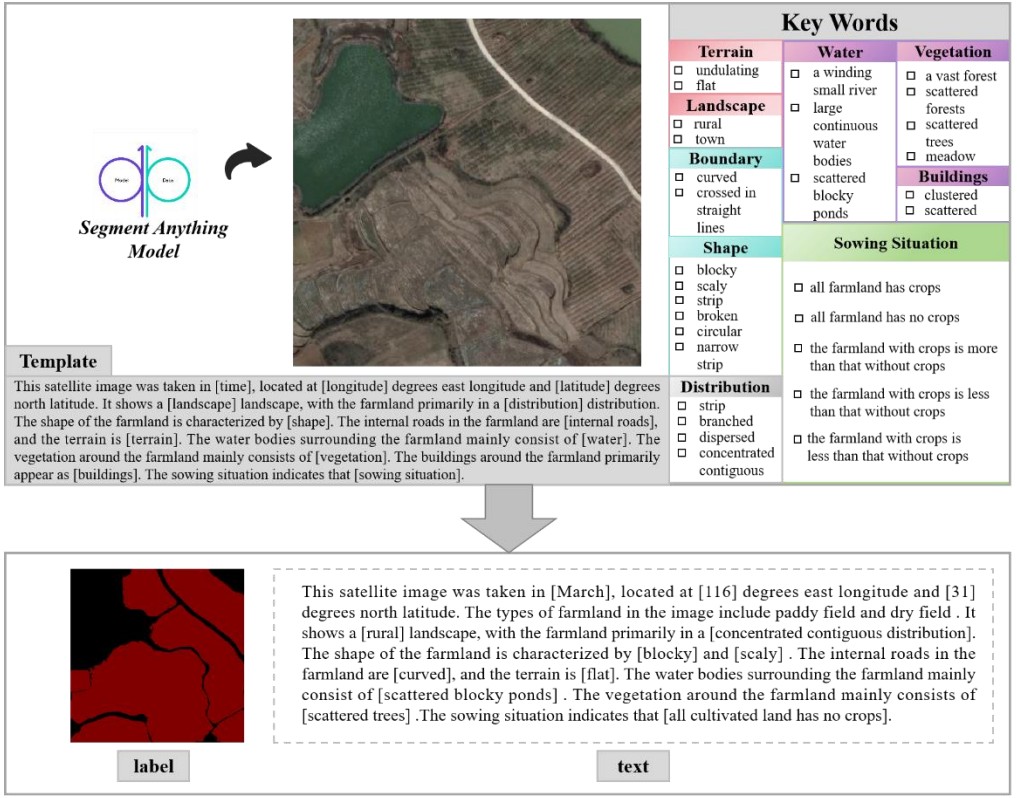

**Fig. 5. Farmland semi-automated annotation framework.**

The semi-automated annotation framework is illustrated in Fig. 5. Specifically, based on keywords related to farmland descriptions, this study first developed a set of farmland caption templates, providing a standardized reference for annotating image samples. To enable semi-automatic text annotation, this study integrated the constructed farmland caption templates and corresponding keywords into the open-source annotation software LabelMe. In this way, when annotating the remote sensing images of farmland, semi-automatic text annotation can be completed by visually observing the visual features of the remote sensing images and combining them with manually selected summarized keywords. In particular, the shooting month and longitude and latitude data of the farmland remote sensing images are automatically extracted from the original data. In addition, due to the limitation of cropping size, some images may not contain any land object categories other than farmland. Therefore, when annotating the surrounding environmental attributes using the semi-automated framework, this study requires that the presence of relevant land cover types be verified first, to ensure the accuracy of the captions. Finally, in order to quickly and accurately obtain high-quality farmland masks, this paper connects the Segment Anything Model (SAM) to LabelMe and performs semi-automatic mask annotation on the image to obtain the image label. Through semi-automatic annotation, humans only need to correct and verify part of the results, which significantly reduces the manpower and time costs compared to traditional fully manual annotation methods. At the same time, the semi-automated process combines the consistency of algorithms with the precision of manual verification, effectively minimizing subjective errors that can occur in manual



annotation and thereby enhancing the accuracy and reliability of the labels.

## 3.2 The Spatiotemporal Characteristics Analysis of FarmSeg-VL Based on Multidimensional Statistics

FarmSeg-VL, as the first large-scale farmland image-text dataset covering multiple regions and seasons in China, is valuable for reflecting the dynamic characteristics of geographical zoning differences, crop growth cycle variations, and tillage practices. This section uses multidimensional statistical methods to analyze the ability of FarmSeg-VL to collaboratively represent spatial breadth and temporal continuity, providing a theoretical basis for evaluating its applicability in cross-regional and cross-seasonal farmland segmentation.

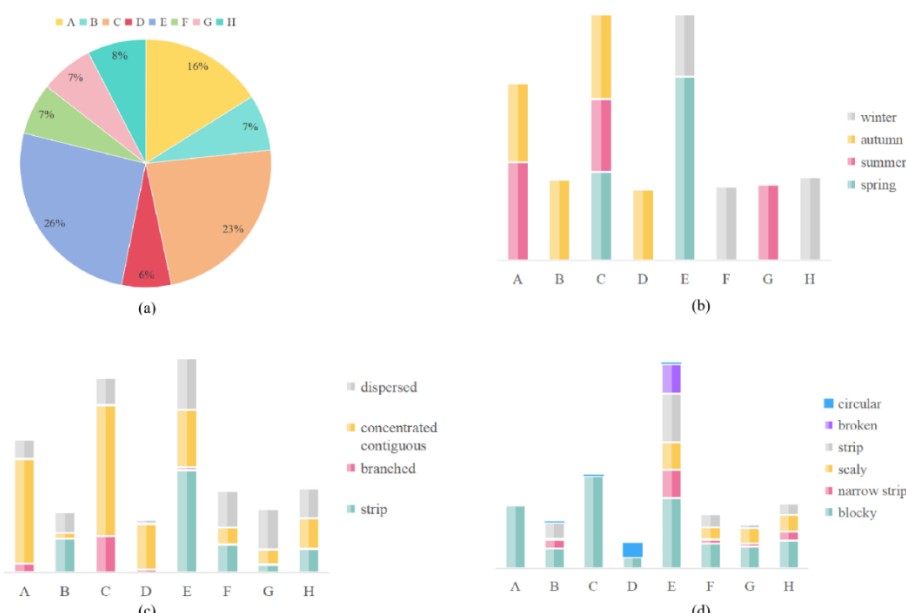

**Fig. 6. Diversity of data samples. (a) Sample distribution ratio across different agricultural regions. (b) Sample distribution ratio for different seasons in each agricultural region. (c) Sample distribution ratio based on different farmland distribution patterns in each agricultural region. (d) Sample distribution ratio for different farmland shapes in each agricultural region. Where A represents the Northeast China Plain, B represents the Northern Arid and Semi-arid Region, C represents the Huang-Huai-Hai Plain, D represents the Loess Plateau, E represents the Yangtze River Middle and Lower Reaches Plain, F represents South China Areas, G represents the Sichuan Basin, and H represents the Yungui Plateau.**

Fig. 6 reveals the spatiotemporal characteristics of FarmSeg-VL from both spatial and temporal perspectives. In terms of the spatial dimension, the sample distribution of agricultural areas in Fig.6(a) shows that FarmSeg-VL fully covers eight agricultural areas, ranging from the Northeast Plain to the Southwest Mountains. Notably, the sample count in the Yangtze River Middle and Lower Reaches Plain is significantly more than in other regions, accurately reflecting the geographical characteristics of the area, which is marked by a high degree of farmland fragmentation and notable terrain complexity. In terms of the temporal dimension, the seasonal distribution in Fig.6 (b) shows that samples in the northern agricultural regions are concentrated in summer and autumn, while the southern agricultural regions exhibit a more balanced distribution



throughout the year. This pattern is closely aligned with the differences in crop growth cycles driven by latitude gradients in China. In addition, Fig.6 (c) and (d) illustrate the distribution patterns and shape characteristics of farmland across eight agricultural regions, highlighting the variations between them. Among these, the agricultural areas in the Yangtze River Middle and Lower Reaches Plain exhibit the greatest diversity, featuring four distinct distribution patterns and six different shape characteristics of farmland. In the Northeast China Plain and the Huang-Huai-Hai Plain, farmland is primarily distributed in

concentrated areas, with a predominantly blocky form. In other agricultural regions, there is a clear correlation between the distribution patterns and the shape characteristics of farmland. The diversity and richness of farmland samples across different agricultural regions fully reflect the spatiotemporal variability captured by FarmSeg-VL, underscoring its advantages in farmland segmentation.

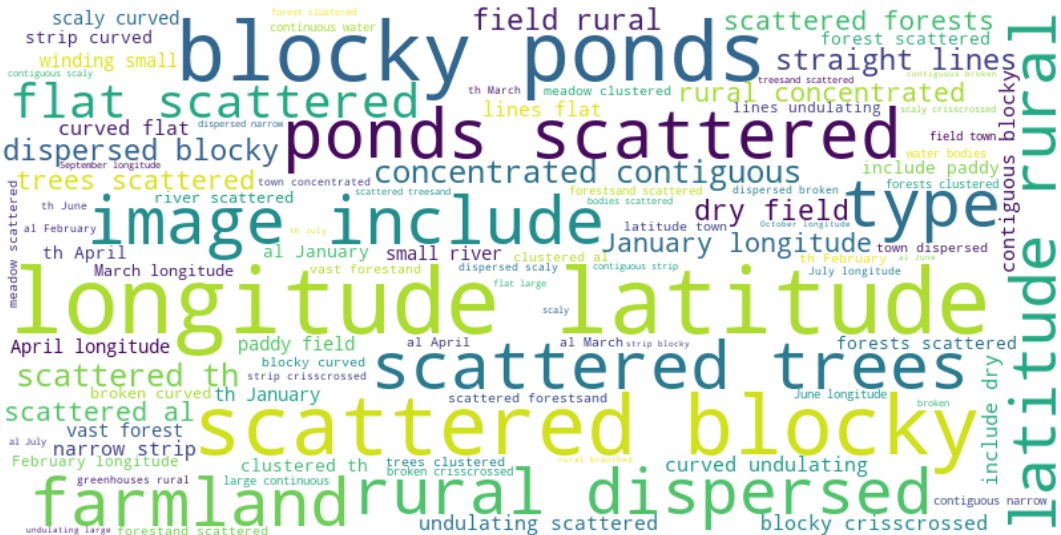

**Fig. 7. Word cloud of farmland captions.**


To further reveal the spatiotemporal characteristics of FarmSeg VL, we extracted keywords from its caption and generated a word cloud of farmland-related attributes. As shown in Fig. 7, the spatiotemporal characteristics of FarmSeg-VL are further illustrated through the keyword cloud. High-frequency spatial attributes (e.g., " latitude" and " longitude") show strong semantic associations with temporal attributes (e.g., "January"), indicating that the captions in the FarmSeg-VL dataset

effectively link temporal and spatial concepts. The spatial differentiation of morphological descriptors such as "concentrated contiguous" and "dispersed" aligns closely with the statistical results shown in Fig. 6(c) and Fig. 6(d), indicating that text annotations can effectively reflect and convey the geographical patterns of farmland morphology. Notably, the prominent presence of non-farmland attributes such as "ponds" and "forests" among the keywords suggests that FarmSeg-VL reflect not only the characteristics of farmland itself but also emphasizes the logical connections between farmland and its surrounding

environment. In summary, the composite captions in FarmSeg-VL at both temporal and spatial levels not only reflect the fundamental characteristics of farmland but also reveal the external driving factors behind its spatiotemporal evolution.



### 3.3 Why is FarmSeg-VL More Suitable as a Dataset Benchmark for Farmland Segmentation?

**Comprehensive spatiotemporal coverage with rich seasonal and regional diversity.** The FarmSeg-VL offers extensive coverage across both temporal and spatial dimensions, spanning all four seasons—spring, summer, autumn, and winter—while also including eight typical agricultural regions of China. The dataset reflects the seasonal differences in agricultural landscapes, as well as the unique geographic features of each region, such as variations in farmland characteristics and surrounding environments. These factors enhance the diversity of the dataset.

**Rich semantic captions capturing comprehensive farmland attributes.** Unlike traditional datasets with simple image annotations, FarmSeg-VL incorporates detailed language captions summarizing the spatiotemporal features of farmland images. Specifically, it covers 11 key descriptive points, including farmland inherent properties, phenological characteristics, spatial distribution, topographic and geomorphic features, and the distribution of surroundings. The rich semantic captions significantly enhance the model's accuracy in farmland segmentation.

**Comprehensive seasonal-regional coverage enhances model robustness.** Seasonal and climatic variations significantly influence farmland morphology and distribution. Unlike traditional datasets, which typically focus on a single season and limit model adaptability, the FarmSeg-VL spans all four seasons, enabling models to better capture seasonal dynamics and varying crop growth conditions. Additionally, FarmSeg-VL covers diverse agricultural regions across China, reflecting distinct differences in farmland characteristics due to climate and geographic variation. The dataset's extensive seasonal and regional coverage enhances the model's robustness, ensuring accurate and efficient farmland segmentation under diverse seasonal and climatic conditions.

### 4 Experiments

This chapter outlines the experimental setup in Section 4.1. Section 4.2 evaluates the effectiveness of the FarmSeg-VL for farmland segmentation by comparing a model fine-tuned on FarmSeg-VL with a vision language model (VLM) trained on a general image-text dataset. This comparison aims to verify whether a dedicated farmland image-text dataset can enhance model performance in farmland segmentation. In Section 4.3, we assess segmentation performance across different agricultural regions, comparing VLMs trained on FarmSeg-VL with the deep learning models that rely solely on labels, including U-Net, DeepLabV3, FCN, and SegFormer. We also analyze the generalization capability of models trained on FarmSeg-VL in diverse agricultural landscapes and their adaptability to spatiotemporal heterogeneity. Section 4.4 investigates the transferability of VLMs trained on FarmSeg-VL through comparative experiments with traditional models on public datasets, evaluating their cross-dataset generalization and cross-domain potential. Finally, Section 4.5 compares FarmSeg-VL with existing farmland datasets in the context of farmland segmentation applications.





### 4.1 Experimental Setup

**Dataset Partitioning.** To avoid the influence of sample similarity between the training, testing, and validation sets on the reliable evaluation of the model's generalization ability and domain transferability, this paper selects samples from different agricultural regions for each set. This approach helps reduce spatial homogeneity and ensures a more robust assessment of the

model's performance. The dataset is divided into training, validation, and test sets in a 7:2:1 ratio. Specifically, the training set comprises 15,821 samples, the validation set contains 4,512 samples, and the test set includes 2,272 samples. The distribution of test set samples across different agricultural regions is as follows: 363 samples from the Northeast China Plain, 531 samples from the Huang-Huai-Hai Plain, 146 samples from the Northern Arid and Semi-Arid Region, 16 samples from the Loess Plateau, 587 samples from the Yangtze River Middle and Lower Reaches Plain, 152 samples from South China, 156 samples

from the Sichuan Basin, and 171 samples from the Yungui Plateau.

**Evaluation Metrics.** To assess model performance, this study uses four widely adopted metrics in farmland segmentation: Mean Accuracy (mACC), Mean Intersection over Union (mIoU), Mean Dice Coefficient (mDice), and Recall. Specifically, mACC represents the average pixel classification accuracy across all categories, while mIoU quantifies the mean ratio of intersection over union, a standard metric in semantic segmentation. mDice measures the similarity between predicted and

ground-truth segmentation results, and Recall evaluates the proportion of correctly identified positive samples, reflecting the model's ability to capture relevant farmland regions.

### 4.2 Fine-Tuning General VLMs with FarmSeg-VL: Bridging Domain Gaps and Enhancing Semantic Comprehension for Farmland Segmentation

In order to verify the advantages of the model trained on FarmSeg-VL in farmland segmentation compared to models trained

on general image-text datasets. This study systematically evaluates the impact of FarmSeg-VL based fine-tuning on farmland segmentation accuracy across three mainstream vision language segmentation models: LISA (Lai et al., 2023), PixelLM(Ren et al., 2024), and LaSagna(Wei et al., 2024). Among them, LISA is a model that integrates a large language model (LLM) with segmentation mask generation capabilities, enabling reasoning-driven segmentation based on complex textual prompts. LaSagnA extends LISA's architecture by adopting a unified sequence format to handle more complex queries while enhancing

perceptual ability through the incorporation of semantic segmentation. This design demonstrates superior performance in processing intricate prompts and improving reasoning capability. PixelLM, in contrast, is a multimodal model specialized for pixel-level reasoning. It addresses the challenge of generating pixel-wise masks for multiple objects by introducing a lightweight pixel decoder and a segmentation codebook, which improves both efficiency and granularity in segmentation tasks.

The experimental results are shown in Table 3. It can be clearly seen that in farmland segmentation, after fine-tuning the

model using the FarmSeg-VL, the performance of the model has been significantly improved, with an improvement of nearly 30% to 40%. Specifically, across all methods, the fine-tuned models consistently achieve higher mIoU scores compared to their non-fine-tuned counterparts, highlighting the effectiveness of FarmSeg-VL in improving segmentation accuracy. This result demonstrates that fine-tuning significantly enhances the model's ability to capture and accurately segment relevant




features. Notably, the PixelLM model does not produce results in its non-fine-tuned state, as it has not been exposed to

farmland-related semantic information during pretraining and is therefore incapable of generating effective predictions without fine-tuning. However, after being trained on the FarmSeg-VL, PixelLM becomes capable of accurately predicting farmland, with performance approaching that of the other two VLMs. This further underscores the importance of fine-tuning with a domain-dedicated dataset to enhance model performance for specialized tasks. To more intuitively analyze the experimental results, this study visualized the segmentation outcomes. As shown in Fig. 8, models that have not undergone fine-tuning tend

to misclassify large areas of buildings and forests as farmland. This suggests that non-fine-tuned models struggle to accurately capture inherent properties of farmland, leading to high uncertainty and significant errors in segmentation results, as well as a lack of stability and consistency.

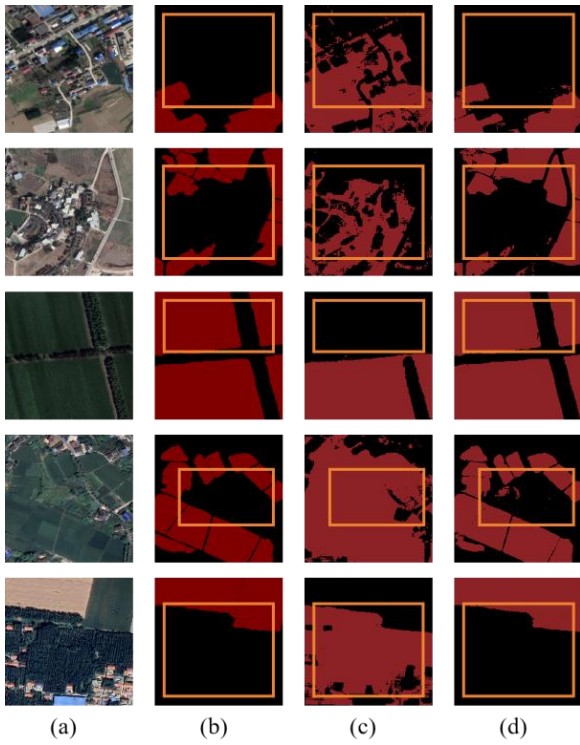

(a)             (b)             (c)             (d)

**Fig. 8. Visualization of partial experimental results fine-tuned on the FarmSeg-VL Dataset. (a) Original image. (b) Ground truth.**
**(c) Test results without fine-tuning. (d) Test results after fine-tuning.**

**Table 3. Comparison of fine-tuning results on the FarmSeg-VL dataset.**

| Method | No Fine Tuning(%) | | | | Fine Tuning(%) | | | |
|---|---|---|---|---|---|---|---|---|
| | mIoU | mACC | mDice | Recall | mIoU | mACC | mDice | Recall |
| LISA | 46.50 | 58.42 | 58.39 | 58.76 | 87.71 | 93.47 | 93.45 | 93.46 |
| PixelLM | / | / | / | / | 83.65 | 91.13 | 91.09 | 91.16 |
| LaSagna | 32.31 | 52.00 | 47.16 | 56.51 | 86.95 | 93.03 | 93.02 | 93.00 |





In summary, the FarmSeg-VL offers more precise domain-dedicated knowledge for farmland segmentation, allowing models to better capture fine-grained features of farmland. Specifically, FarmSeg-VL contains high-quality farmland annotations that cover multiple semantic dimensions, such as farmland shape, distribution, and sowing situation. This comprehensive information significantly improves the model's ability to understand and segment farmland features with greater accuracy. Compared to general datasets, FarmSeg-VL effectively reduces cross-domain discrepancies, allowing the model to focus on farmland features, thereby further enhancing the accuracy of farmland segmentation.

**4.3 Comparing Model Performance Trained on FarmSeg-VL in Different Agricultural Regions**

To explore the application effect of models trained on the FarmSeg VL in different agricultural regions, this section divides the test set into various agricultural regions, including the Northeast China Plain, Huang-Huai-Hai Plain, Northern Arid and Semi-Arid Region, Loess Plateau, Yangtze River Middle and Lower Reaches Plain, South China, Sichuan Basin, and Yungui Plateau. These regions will be tested using both vision-language models (PixelLM, LaSagna, LISA) and the deep learning models that rely solely on labels (U-Net, DeepLabV3, FCN, SegFormer). Notably, these models that rely solely on labels do not incorporate any language modality, they are trained and tested exclusively using original farmland image and ground truth.

Tables 4 to 11 display the testing accuracy of the model in different agricultural regions. From the overall results, both the deep learning models that rely solely on labels and VLMs demonstrated strong testing accuracy in the agricultural regions of the Northeast China Plain and the Huang-Huai-Hai Plain. However, in the remaining six agricultural regions, the performance differences between the two model types became more pronounced. The primary reason for these differences lies in the varying complexity of the spatial structure of farmland across different agricultural regions. In the Northeast China Plain and Huang-Huai-Hai Plain, the terrain is relatively flat, and the farmland is distributed in a more regular and contiguous manner. As a result, both models exhibit strong segmentation performance in these relatively simple scenarios. In other agricultural regions, particularly in South China Areas, the farmland generally exhibits scattered and fragmented characteristics. Additionally, it shares a high degree of textural similarity with surrounding non-farmland features, such as forests and water bodies, which makes it difficult for the model to segment farmland. By incorporating language, VLMs can effectively comprehend the spatial distribution of farmland and its surrounding environment, thereby alleviating the segmentation challenges caused by spatial differentiation and demonstrating advantages in these different agricultural regions.

To visually illustrate the performance differences among various models in farmland segmentation tasks, Figures 9 to 16 present the segmentation results for each agricultural region. From this, it can be observed that in agricultural regions such as the Northeast China Plain and the Huang-Huai-Hai Plain, although the overall accuracy is high, the deep learning models that rely solely on labels still exhibit certain limitations. For example, this type of model is prone to misjudgment when encountering terrain features that resemble farmland, such as ponds and grasslands, and often exhibits issues such as boundary blurring and discontinuity in the segmentation of farmland. In South China Areas, the highly fragmented nature of farmland, with its scattered or narrow distribution, the segmentation challenge is further acerbated. The deep learning models that rely solely on labels struggle to effectively identify such atypical farmland, leading to a significant decrease in segmentation



accuracy. In contrast, VLMs have demonstrated notable advantages in the aforementioned agricultural regions. By
incorporating farmland-related key words—such as "concentrated buildings" and "narrow strips", VLMs enhance their
comprehension of both the inherent properties of farmland and the contextual information of its surrounding environment.
This enriched understanding contributes to improved completeness and accuracy in farmland segmentation. In addition, this
advantage is not limited to the aforementioned agricultural regions but is also consistently performance in the segmentation
results in the other five regions. This further validates the generalization capability and robustness of the VLMs in diverse
agricultural landscapes.

In summary, compared to the deep learning models that rely solely on labels, VLMs that incorporate caption demonstrate
significant advantages in farmland segmentation across all agricultural regions. Language information effectively compensates
for the limitations of the deep learning models that rely solely on labels in complex scenarios, enhancing the model's
understanding of farmland morphology and the relationship between farmland and surrounding land cover, thereby
significantly improving farmland segmentation accuracy.

Table 4. Farmland segmentation results of different methods in Northeast China Plain.

| Evaluation Metrics(%) | The deep learning models that rely solely on labels | | | | Vision-Language Model | | |
|---|---|---|---|---|---|---|---|
| | U-Net | Deeplabv3 | FCN | SegFormer | PixelLM | LaSagna | LISA |
| mACC | 82.56 | 86.75 | 91.22 | 91.03 | 93.16 | 94.85 | **95.15** |
| mIoU | 73.52 | 78.60 | 84.70 | 84.91 | 85.88 | 89.15 | **89.75** |
| mDice | 84.40 | 87.84 | 91.64 | 91.76 | 92.35 | 94.23 | **94.57** |
| Recall | 82.56 | 86.75 | 91.22 | 91.03 | 92.32 | 94.29 | **94.56** |

Table 5. Farmland segmentation results of different methods in Huang-Huai-Hai Plain.

| Evaluation Metrics(%) | The deep learning models that rely solely on labels | | | | Vision-Language Model | | |
|---|---|---|---|---|---|---|---|
| | U-Net | Deeplabv3 | FCN | SegFormer | PixelLM | LaSagna | LISA |
| mACC | 91.38 | 91.71 | 94.37 | 94.32 | 94.11 | 95.51 | **95.97** |
| mIoU | 85.53 | 84.56 | 88.35 | 89.59 | 88.11 | 90.79 | **91.70** |
| mDice | 92.15 | 91.59 | 93.79 | 94.49 | 93.65 | 95.16 | **95.66** |
| Recall | 91.38 | 91.71 | 94.37 | 94.32 | 93.79 | 95.27 | **95.72** |

Table 6. Farmland segmentation results of different methods in Northern Arid and Semi-arid Region.

| Evaluation Metrics(%) | The deep learning models that rely solely on labels | | | | Vision-Language Model | | |
|---|---|---|---|---|---|---|---|
| | U-Net | Deeplabv3 | FCN | SegFormer | PixelLM | LaSagna | LISA |
| mACC | 81.11 | 82.59 | 82.67 | 86.91 | 88.37 | **90.74** | 90.53 |
| mIoU | 68.30 | 70.46 | 70.40 | 76.97 | 79.14 | **83.02** | 82.70 |
| mDice | 81.15 | 82.64 | 82.63 | 86.98 | 88.36 | **90.72** | 90.53 |
| Recall | 81.11 | 82.59 | 82.67 | 86.91 | 88.39 | **90.77** | 90.52 |




**Table 7. Farmland segmentation results of different methods in Loess Plateau.**

| Evaluation | The deep learning models that rely solely on labels | | | | Vision-Language Model | | |
|---|---|---|---|---|---|---|---|
| Metrics(%) | U-Net | Deeplabv3 | FCN | SegFormer | PixelLM | LaSagna | LISA |
| mACC | 74.88 | 83.07 | 87.24 | 93.02 | 92.77 | 95.11 | **95.74** |
| mIoU | 58.01 | 70.74 | 77.23 | 87.13 | 86.50 | 90.68 | **91.82** |
| mDice | 73.23 | 82.86 | 87.15 | 93.12 | 92.76 | 95.11 | **95.73** |
| Recall | 74.88 | 83.07 | 87.24 | 93.02 | 92.76 | 95.11 | **95.78** |

**Table 8. Farmland segmentation results of different methods in Yangtze River Middle and Lower Reaches Plain.**

| Evaluation | The deep learning models that rely solely on labels | | | | Vision-Language Model | | |
|---|---|---|---|---|---|---|---|
| Metrics(%) | U-Net | Deeplabv3 | FCN | SegFormer | PixelLM | LaSagna | LISA |
| mACC | 84.62 | 88.59 | 89.57 | 89.53 | 90.20 | 91.53 | **92.07** |
| mIoU | 72.26 | 78.27 | 80.06 | 80.08 | 80.82 | 83.22 | **84.14** |
| mDice | 83.75 | 87.72 | 88.85 | 88.86 | 89.31 | 90.79 | **91.33** |
| Recall | 84.26 | 88.59 | 89.57 | 88.35 | 89.28 | 90.64 | **91.39** |

**Table 9. Farmland segmentation results of different methods in South China Areas.**

| Evaluation | The deep learning models that rely solely on labels | | | | Vision-Language Model | | |
|---|---|---|---|---|---|---|---|
| Metrics(%) | U-Net | Deeplabv3 | FCN | SegFormer | PixelLM | LaSagna | LISA |
| mACC | 65.86 | 71.85 | 79.74 | 71.64 | 89.89 | 91.27 | **91.48** |
| mIoU | 53.09 | 62.29 | 67.37 | 63.20 | 71.36 | 74.07 | **74.52** |
| mDice | 65.57 | 74.13 | 78.95 | 74.83 | 82.10 | 84.13 | **84.45** |
| Recall | 65.86 | 71.85 | 79.74 | 71.64 | 81.84 | 84.80 | **85.23** |

**Table 10. Farmland segmentation results of different methods in Sichuan Basin.**

| Evaluation | The deep learning models that rely solely on labels | | | | Vision-Language Model | | |
|---|---|---|---|---|---|---|---|
| Metrics(%) | U-Net | Deeplabv3 | FCN | SegFormer | PixelLM | LaSagna | LISA |
| mACC | 87.61 | 89.68 | 91.50 | 91.46 | 93.14 | 93.66 | **94.18** |
| mIoU | 72.87 | 76.82 | 82.45 | 82.21 | 84.45 | 85.51 | **86.52** |
| mDice | 84.02 | 86.66 | 90.22 | 90.08 | 91.43 | 92.07 | **92.67** |
| Recall | 87.61 | 89.68 | 91.50 | 91.46 | 91.24 | 91.89 | **92.85** |


**Table 11. Farmland segmentation results of different methods in Yungui Plateau.**

| Evaluation | The deep learning models that rely solely on labels | | | | Vision-Language Model | | |
|---|---|---|---|---|---|---|---|
| Metrics(%) | U-Net | Deeplabv3 | FCN | SegFormer | PixelLM | LaSagna | LISA |
| mACC | 76.47 | 82.82 | 84.98 | 85.96 | 87.18 | 89.04 | **90.11** |
| mIoU | 62.95 | 71.50 | 74.51 | 75.28 | 76.52 | 79.62 | **81.44** |
| mDice | 77.00 | 83.25 | 85.30 | 85.84 | 86.64 | 88.61 | **89.73** |
| Recall | 76.47 | 82.82 | 84.98 | 85.96 | 86.76 | 88.61 | **89.69** |



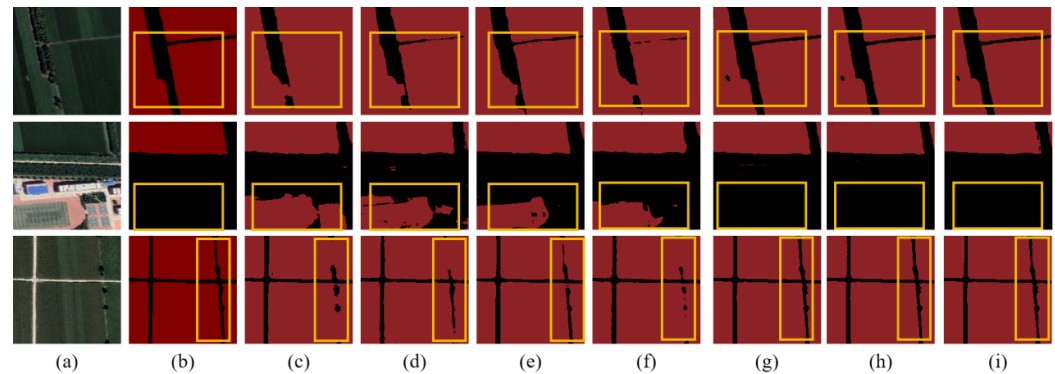

**Fig. 9. Farmland segmentation results of different methods in Northeast China Plain. (a)Original image. (b)groundtruth. (c)U-Net. (d)Deeplabv3. (e)FCN. (f)SegFormer. (g)PixelLM. (h)LaSagna. and (i)LISA.**

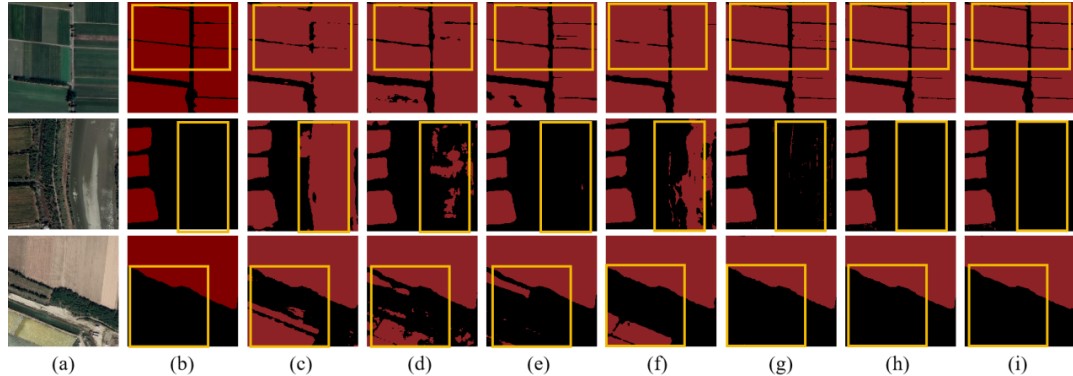


**Fig. 10. Farmland segmentation results of different methods in Huang-Huai-Hai Plain . (a)Original image. (b)groundtruth. (c)U-Net. (d)Deeplabv3. (e)FCN. (f)SegFormer. (g)PixelLM. (h)LaSagna. and (i)LISA.**

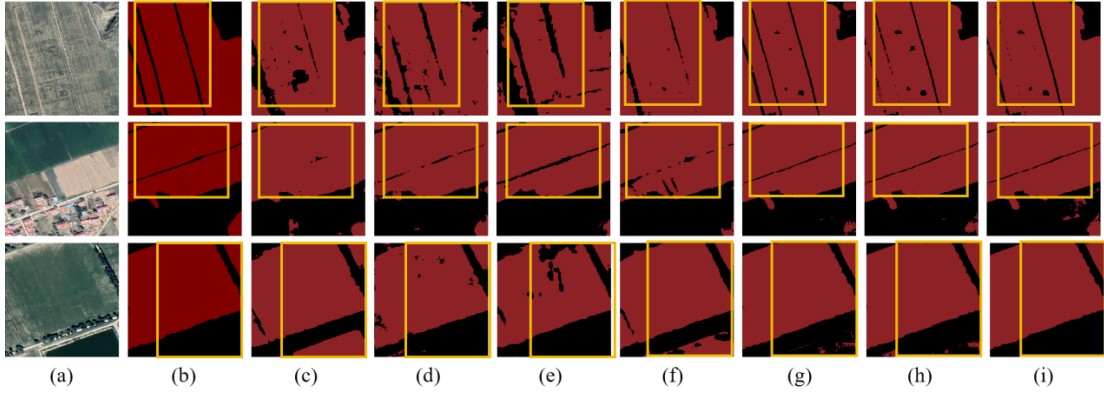

**Fig. 11. Farmland segmentation results of different methods in Northern Arid and Semi-arid Region . (a)Original image. (b)groundtruth. (c)U-Net. (d)Deeplabv3. (e)FCN. (f)SegFormer. (g)PixelLM. (h)LaSagna. and (i)LISA.**




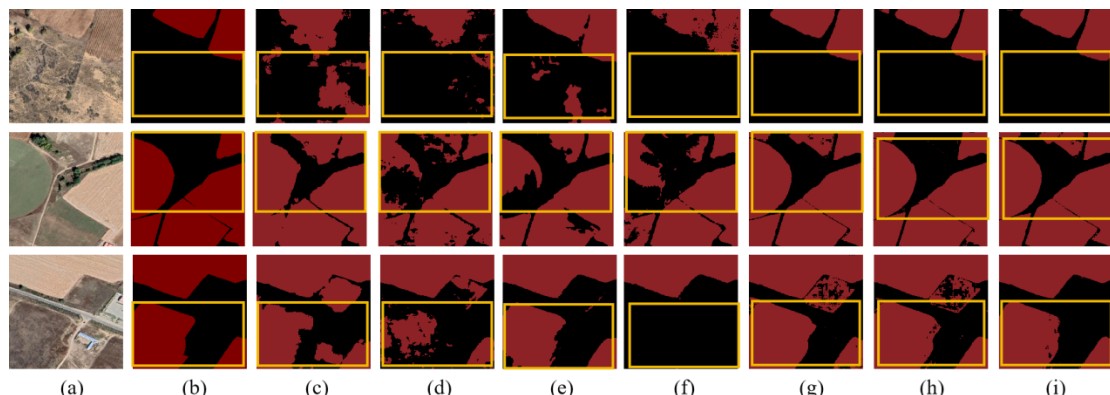

**Fig. 12. Farmland segmentation results of different methods in Loess Plateau . (a)Original image. (b)groundtruth. (c)U-Net. (d)Deeplabv3. (e)FCN. (f)SegFormer. (g)PixelLM. (h)LaSagna. and (i)LISA.**

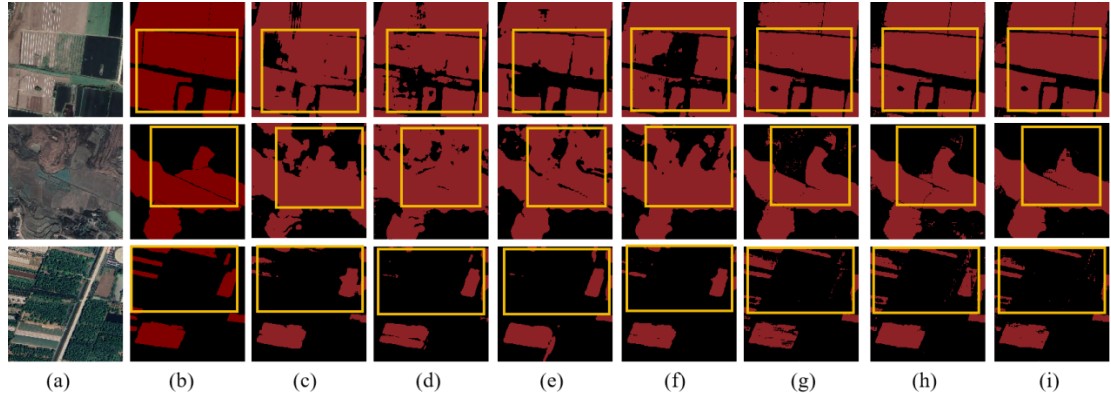

**Fig. 13. Farmland segmentation results of different methods in Yangtze River Middle and Lower Reaches Plain . (a)Original image. (b)groundtruth. (c)U-Net. (d)Deeplabv3. (e)FCN. (f)SegFormer. (g)PixelLM. (h)LaSagna. and (i)LISA.**

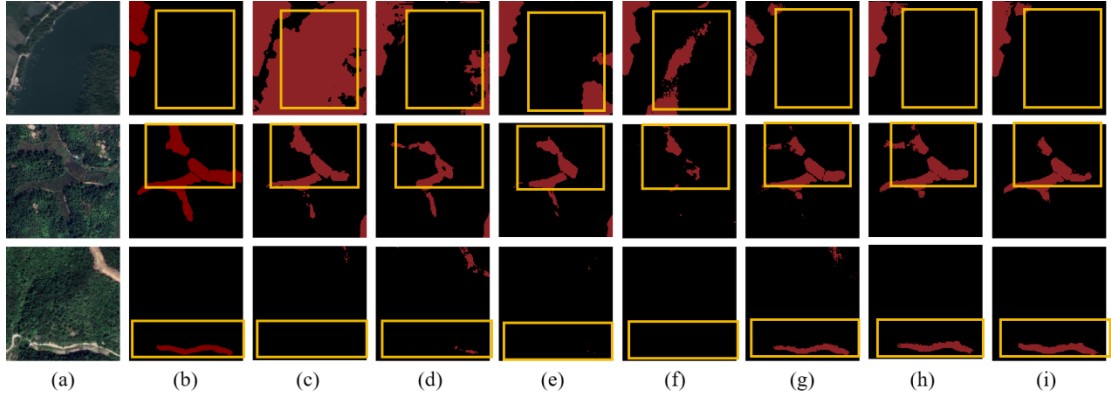

**Fig. 14 Farmland segmentation results of different methods in South China Areas . (a)Original image. (b)groundtruth. (c)U-Net. (d)Deeplabv3. (e)FCN. (f)SegFormer. (g)PixelLM. (h)LaSagna. and (i)LISA.**





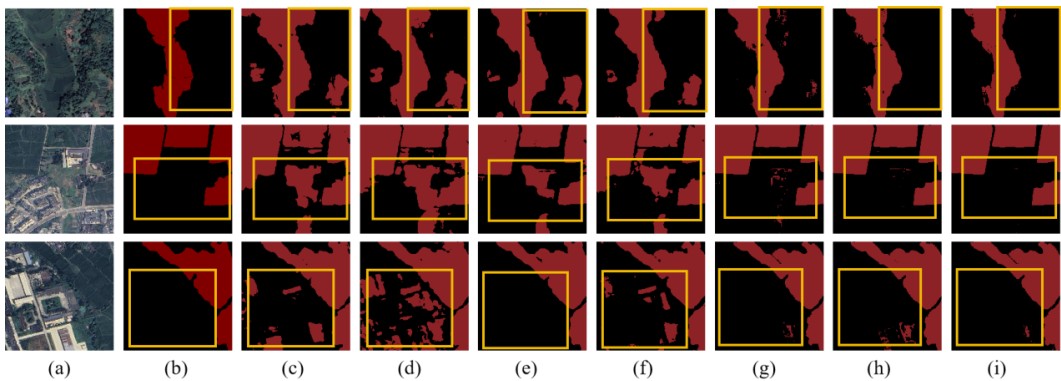


**Fig. 15 Farmland segmentation results of different methods in Sichuan Basin . (a)Original image. (b)groundtruth. (c)U-Net. (d)Deeplabv3. (e)FCN. (f)SegFormer. (g)PixelLM. (h)LaSagna. and (i)LISA.**

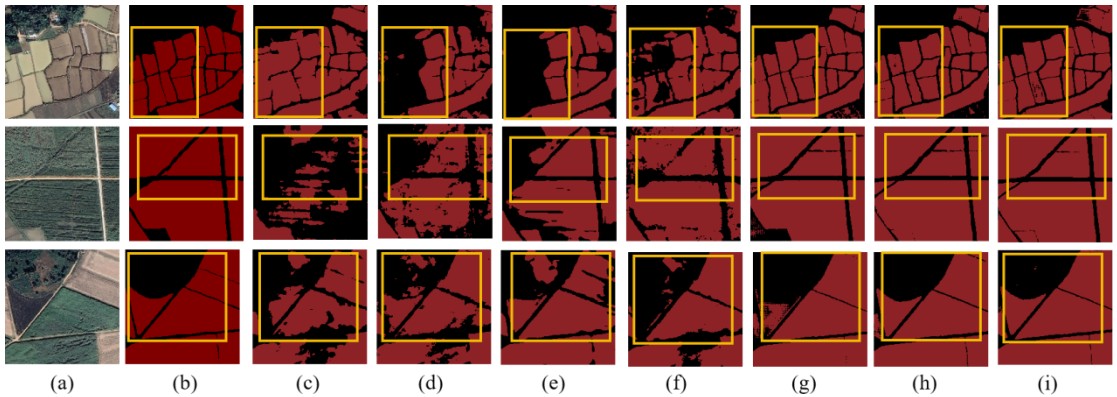

**Fig. 16 Farmland segmentation results of different methods in Yungui Plateau . (a)Original image. (b)groundtruth. (c)U-Net.**
**(d)Deeplabv3. (e)FCN. (f)SegFormer. (g)PixelLM. (h)LaSagna. and (i)LISA.**

## 4.4 Cross-Domain Performance Evaluation of Models Trained on FarmSeg-VL

In order to evaluate the performance of models trained on the FarmSeg-VL dataset in cross-domain tasks, this paper conducted relevant experiments. Specifically, this section presents transfer tests using VLMs (PixelLM, LaSagna, LISA) and the deep learning models that rely solely on labels (U-Net, DeepLabV3, FCN, SegFormer) trained on the FarmSeg-VL across multiple
public datasets. The test datasets include DeepGlobe Land Cover (DGLC), LoveDA, and the Fine-Grained Farmland Dataset (FGFD). Specifically, the DGLC dataset covers regions in Thailand, Indonesia, and India, while the LoveDA includes areas in Nanjing, Changzhou, and Wuhan in China. The FGFD farmland dataset encompasses regions such as Heilongjiang, Hebei, Shaanxi, Guizhou, Hubei, Jiangxi, and Tibet in China. The specific details are provided in Table 1. Specifically, to maintain consistency with the FarmSeg-VL test set and ensure the data is more suitable for the model, we performed data preprocessing
on the DGLC and LoveDA. This preprocessing primarily involved cropping the images to a size of 512×512 and merging non-farmland pixel labels, among other steps.





**Table 12. Farmland segmentation results of different methods on FGFD.**

| Evaluation | The deep learning models that rely solely on labels | | | | Vision-Language Model | | |
|---|---|---|---|---|---|---|---|
| Metrics(%) | U-Net | Deeplabv3 | FCN | SegFormer | PixelLM | LaSagna | LISA |
| mACC | 72.38 | 74.76 | 76.52 | 76.40 | 78.59 | 80.70 | **83.33** |
| mIoU | 57.48 | 60.11 | 62.43 | 62.34 | 64.68 | 66.83 | **70.58** |
| mDice | 72.71 | 74.94 | 76.74 | 76.66 | 78.55 | 80.00 | **82.65** |
| Recall | 72.38 | 74.76 | 76.52 | 76.40 | 78.98 | 80.84 | **83.87** |

**Table 13. Farmland segmentation results of different methods on LoveDA.**

| Evaluation | The deep learning models that rely solely on labels | | | | Vision-Language Model | | |
|---|---|---|---|---|---|---|---|
| Metrics(%) | U-Net | Deeplabv3 | FCN | SegFormer | PixelLM | LaSagna | LISA |
| mACC | 70.83 | 77.05 | 73.65 | 73.78 | 78.79 | 80.45 | **81.76** |
| mIoU | 47.77 | 63.85 | 61.41 | 60.57 | 60.75 | 64.03 | **65.74** |
| mDice | 64.65 | 77.47 | 75.22 | 74.73 | 74.78 | 77.54 | **78.82** |
| Recall | 70.83 | 77.05 | 73.65 | 73.78 | 77.73 | 78.87 | **80.75** |

**Table 14. Farmland segmentation results of different methods on DGLC.**

| Evaluation | The deep learning models that rely solely on labels | | | | Vision-Language Model | | |
|---|---|---|---|---|---|---|---|
| Metrics(%) | U-Net | Deeplabv3 | FCN | SegFormer | PixelLM | LaSagna | LISA |
| mACC | 64.60 | 71.73 | 69.10 | 70.32 | 66.13 | 71.69 | **72.23** |
| mIoU | 48.73 | 55.68 | 50.17 | 52.15 | 49.38 | 55.78 | **56.36** |
| mDice | 64.71 | 71.41 | 66.81 | 68.55 | 66.11 | 71.59 | **72.06** |
| Recall | 64.60 | 71.73 | 69.10 | 70.32 | 69.14 | 72.22 | **72.44** |

Tables 12-14 present the experimental results on the FGFD, LoveDA, and DGLC, respectively. Overall, both the deep learning models that rely solely on labels and VLMs exhibit strong cross-domain transfer transferability. This can be attributed to the FarmSeg-VL dataset's broad geographic coverage and diverse seasonal variations, which provide a solid foundation for cross-domain feature learning. Notably, VLMs demonstrate significantly superior cross-domain transfer performance across all three datasets compared to traditional labeled data-dependent deep learning models. This advantage is primarily attributed

to the fine-grained captions provided by FarmSeg-VL, which inject transferable semantic prior knowledge into the VLMs. For instance, when caption prompts such as "strip-shaped farmlands in spring" are provided, the models autonomously correlate farmland shape characteristics across different regions under spring conditions. This integration of semantic priors enables VLMs to overcome the representational limitations inherent in single-modality visual features, thereby maintaining enhanced discriminative capabilities in cross-domain scenarios.

Through the cross-domain experiments, this study has drawn two key conclusions: Firstly, models trained on the FarmSeg-VL exhibit significant cross-domain transferability, fully demonstrating the improvement of model generalization performance by the FarmSeg-VL. Secondly, the introduction of captions breaks through the limitations of the deep learning models that





rely solely on labels, enabling the model to decouple spatiotemporal heterogeneity interference and effectively improve segmentation accuracy in complex farming scenes.

## 4.5 Enhanced Model Transferability: Comparative Analysis of FarmSeg-VL and Conventional Farmland Datasets

To verify that the model trained on the FarmSeg-VL outperforms models trained on existing farmland datasets in both segmentation accuracy and generalization, we conducted extensive comparative experiments in this section. First, to ensure the reliability of the experimental results, this study uses the latest dedicated dataset, FGFD, as a benchmark for comparison. Since most existing farmland datasets follow the traditional "Image + Label" format (i.e., a paradigm that solely relies on labeled data), four commonly used the deep learning models that rely solely on labels—U-Net, Deeplabv3, FCN, and SegFormer—are selected to train on the FGFD dataset. For the proposed FarmSeg-VL dataset, three VLMs are selected for comparative experiments. Additionally, to ensure fairness, all trained models are uniformly tested on the LoveDA dataset.

The experimental results, shown in Table 15, reveal that models trained on the FarmSeg-VL dataset using VLMs outperform those trained on the FGFD dataset with the deep learning models that rely solely on labels when tested on the LoveDA dataset. Specifically, the mIoU improved by 10% to 40%, and the mAcc increased by 10% to 30%. This gap indicates that models trained on the FarmSeg VL dataset with added language modality have significant transferability in farmland segmentation compared to models trained on the traditional dataset FGFD. Moreover, FarmSeg-VL reflects multiple aspects of farmland characteristics through captions—such as phenological characteristics, spatial distribution, topographic and geomorphic features and distribution of surrounding environments—allowing the model to learn rich and comprehensive information about farmland. With these detailed captions of farmland, models trained on the FarmSeg-VL not only improve the accuracy of farmland segmentation but also enhance the model's ability to handle complex scenes. In summary, the FarmSeg-VL is a large-scale, high-quality image-text dataset of farmland, it has demonstrated great potential in cross-scenario farmland segmentation and provides a strong data foundation for future research in farmland segmentation.

**Table 15. Performance of different datasets and methods on the LoveDA dataset.**

| Evaluation Metrics(%) | FGFD | | | | FarmSeg-VL | | |
|---|---|---|---|---|---|---|---|
| | U-Net | Deeplabv3 | FCN | SegFormer | PixelLM | LaSagna | LISA |
| mACC | 63.80 | 57.93 | 59.19 | 67.48 | 78.79 | 80.45 | **81.76** |
| mIoU | 38.15 | 29.78 | 36.62 | 50.08 | 60.75 | 64.03 | **65.74** |
| mDice | 55.17 | 45.33 | 53.61 | 66.29 | 74.78 | 77.54 | **78.82** |
| Recall | 63.80 | 57.93 | 59.19 | 67.48 | 77.73 | 78.87 | **80.75** |

## 5 Data availability

The FarmSeg-VL dataset is accessible on the Zenodo data repository at https://doi.org/10.5281/zenodo.15099885(Tao et al., 2025).The FarmSeg VL dataset consists of image data, labels, and corresponding farmland text descriptions in JSON files.

## 6 Conclusion

This study constructs FarmSeg-VL, a high-quality image-text dataset specifically designed for farmland segmentation, with
key features including high-precision images and masks, extensive spatiotemporal coverage, and refined captions of farmland
characteristics. In the dataset construction process, Google imagery with a resolution of 0.5-2 meters was selected as the image
data source. Through in-depth analysis of numerous farmland samples, five key attributes were summarized: inherent
properties, phenological characteristics, spatial distribution, topographic and geomorphic features and distribution of
surrounding environments. These were further refined into 11 specific descriptive dimensions, covering shape, boundary
patterns, season, sowing situation, geographic location, distribution, terrain, landscape features, as well as the distribution of
water bodies, buildings, and trees in the surrounding environment. Based on the above keywords, a farmland description
template was designed, and a semi-automated annotation method was used to generate binary mask labels and their
corresponding captions for each image. Ultimately, a dedicated dataset consisting of 22,605 image-text pairs was constructed.
Experimental results show that the model trained on FarmSeg-VL significantly improves accuracy and robustness in farmland
segmentation. As the first large-scale image-text dataset for farmland segmentation, FarmSeg-VL holds significant academic
value and application potential. It is expected to advance research on semantic understanding of farmland in remote sensing
imagery, promote the development of more efficient and generalized segmentation models, and better serve the diverse needs
of agricultural monitoring.

### Author contributions

The dataset was conceptualized by CT and HYW. DDZ, WLM, and ZFD carried out the dataset construction, and related
experiments. DDZ prepared the initial draft of the manuscript, which was reviewed and revised by all authors.

### Competing interests

The contact author has declared that none of the authors has any competing interests.

### Financial support

This research was supported in part by the Natural Science Foundation of Hunan for Distinguished Young Scholars under
Grant 20221110072 and in part by the National Natural Science Foundation of China under Grant 42471419 and Grant
42171376.





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
