# Peer review of "A large-scale image-text dataset benchmark for farmland segmentation"

_Earth System Science Data, 2025_

## Author Response (AR1)

**-Detailed Response to Reviewers-**

Dear Editors and Reviewers:

Thank you for your letter and for your comments concerning our manuscript entitled "A large-scale image-text dataset benchmark for farmland segmentation". These comments are very helpful for improving the quality of our manuscript. Based on your constructive feedback, we have made some corrections and highlighted the response to Reviewer 1 and Reviewer 2 in red font in the revised manuscript. The responses to these comments of reviewers are as follows.

**Responses to the Editor**

Dear Editor,

Thank you for your comments. In this revision, we responded to each reviewer's comments one by one and recorded all the changes in the revised manuscript in detail, sorting them in the order of (1) reviewer comments; (2) author responses; (3) author revisions to the manuscript. Specifically, we made the following changes based on your comments to improve the quality of the manuscript. All the changes in the revised manuscript are as follows:

1. In the introduction, we further supplemented the construction details of the vision language model.

2. A quantitative description of the experimental results was added to the Abstract and Conclusion.

3. In response to Section 3.1, in 1) RS Image Acquisition and Processing and 2) Caption Construction, we elaborated on the considerations of FarmSeg-VL for different agricultural regions in China; at the same time, we added supporting literature for 11 key elements in 2) Caption Construction.

4. New appendices A-F are added: Appendix A supplements more examples of farmland texture descriptions in remote sensing image-text datasets; Appendix B adds supplementary examples of five types of farmland shapes; for the convenience of readers, Tables 4-11 in the text are converted into figures and placed in Appendix C together with Figures 9-16; Appendix D quantitatively compares the time efficiency of traditional manual methods and semi-automated annotation methods; Appendix E supplements the transferability experiments of the LISA model trained on FarmSeg-VL to European countries; Appendix F adds experiments and discussions on the impact of different data distributions on model robustness.

5. We checked the entire article and corrected spelling and grammatical errors.

6. For easier reading, we replaced the figures with more clarity and adjusted the order of the figures and text.

**Point-to-Point responses to the Reviewers.**

**Reviewer 1:**

This paper proposes FarmSeg-VL, the first large-scale image-text benchmark dataset for farmland segmentation, which fills the gap of the lack of high-quality farmland multimodal data in the field of remote sensing. The research has significant innovation and application value, the experimental design is systematic, the results are analyzed in detail, the data are open and transparent, and it meets the publication criteria of journals. However, some of the methodological details, scope of application, and writing expressions need to be further optimized.

**Response:** Thank you for your positive and constructive comments. We sincerely appreciate your recognition of the novelty, application value, and completeness of our work, including the contribution of the FarmSeg-VL dataset and the experimental design. In response to your suggestions regarding methodological details, scope of application, and writing clarity, we have carefully revised the manuscript. Specifically, we have (1) provided further clarification on the methodological design and key parameters, (2) expanded the discussion on the dataset's applicability across different agricultural conditions, and (3) refined the language throughout the manuscript to improve readability and precision. We hope that these revisions address your concerns and further enhance the quality of the paper.

**Point 1:**

The specific application of the model in annotation, such as parameter setting and manual correction ratio, needs to be further explained. In addition, it is necessary to quantify the improvement of annotation efficiency, such as the comparison of time consumption with traditional manual annotation.

**Response 1:**

Thank you for your comment. In response to the specific application of the model in annotation, we provide the following additional explanation. As shown in Fig. 5 of the manuscript, for mask annotation, we integrated the Segment Anything Model (SAM) to assist in generating farmland masks by creating AI-generated polygons. For text annotation, the tool automatically extracts longitude, latitude, and acquisition month information from the image filenames and populates the corresponding fields. Additionally, we designed a farmland keyword selection widget to construct standardized image description texts. The resulting mask and text information are stored in separate JSON files. Regarding the parameter settings, we have used the default parameters of SAM without any additional adjustments. Concerning the manual correction ratio, in most cases, the generated masks require no manual modification, with only approximately 10% of the images necessitating minor boundary adjustments.

[Figure]

**Fig. 5. Farmland semi-automated annotation framework.**

To quantify the improvement in annotation efficiency, we conducted a comparative experiment. Four operators were randomly selected to annotate 13 remote sensing images using both traditional manual tracing and semi-automatic annotation method . The experimental results, shown in Fig. 19 of the revised manuscript, demonstrate that the average annotation time using semi-automatic annotation method was significantly reduced compared to the traditional approach, resulting in a 1.5 times improvement in overall efficiency. This validates the significant improvement in annotation efficiency and usability of the developed tool. The detailed modifications are as follows:

**Appendix**

**D Quantitative evaluation of semi-automated annotation efficiency**

In order to quantify the annotation efficiency of the semi-automatic annotation framework proposed in this article, comparative experiments were conducted in this section. Specifically, we randomly selected four annotators and annotated the masks and texts on 13 farmland remote sensing images using traditional manual drawing methods and semi-automated annotation methods. Finally, we compared the completion time of the annotations. As shown in Fig. 27, after using the semi-automated annotation method, the average annotation time was significantly reduced, saving approximately 2 minutes per image, and overall efficiency improved by 1.5 times. This result indicates that the annotation tool developed in this article has significantly improved efficiency and usability.

[Figure]

**Fig. 27. Comparison of farmland annotation efficiency.**

**Point 2:**

It is suggested to add the selection basis of the 11 key elements, such as whether they have been verified by experts or supported by the literature, in order to enhance the scientific nature of the description framework.

**Response 2:**

Thank you for your comment. Regarding the selection criteria for the 11 key elements, we have added supporting references from the literature. The detailed modifications are as follows:

**3.1 Construction of FarmSeg-VL**

**2) Caption Construction**

For the caption construction of each farmland sample, this study summarizes 11 key elements for describing farmland: shape, boundary morphology, shooting time, sowing conditions, the macro-level distribution of farmland, geographic location information, topographical features, landscape, the distribution of buildings, water bodies, and vegetation. The spatiotemporal characteristics of farmland result from the interaction of multiple factors. (Wang et al., 2022b) Temporally, the variations in crop growth stages lead to distinct visual texture differences in farmland across different seasons. (Zhu et al., 2022) Spatially, farmland exhibits significant spatial differentiation, with different regions affected by factors such as topography, terrain, and water-heat conditions, resulting in noticeable variations in farmland morphology and layout. (Pan and Zhang, 2022) Therefore, this study considers the issue at multiple spatial scales. At the macro-regional scale, typical farmland images were collected from various agricultural regions across China. These regions are not only located in different latitudes and longitudes, but also have different terrains and topography. For instance, farmland in the Northeast China Plain is flat and typically follows a concentrated distribution pattern with regular shapes, which is reflected

in descriptions such as "the farmland primarily exhibits concentrated contiguous distribution" and "the shape of the farmland is characterized by blocky." In contrast, the terrain of South China is predominantly hilly and mountainous, leading to a more dispersed farmland distribution and irregular shapes, which is described in the text as "with the farmland primarily in a dispersed distribution" and "the terrain is undulating." Similarly, farmland in regions like the Loess Plateau and the arid and Semi-Arid Northern Areas often displays terraced or sloping patterns. At the same time, the spatial coupling relationships between farmland and surrounding features, such as water bodies and buildings, are key factors influencing the distribution and accuracy of farmland identification.(Duan et al., 2022; Zheng et al., 2022) The relationship between the farmland and surrounding environmental features is expressed, for example, as "the water bodies surrounding the farmland mainly consist of scattered blocky ponds," and "the vegetation around the farmland mainly consists of scattered trees and scattered forests. " Similarly, the segmentation of farmland relies on boundary and texture information, the shape of the farmland and the boundary morphology, is also crucial for accurate identification of farmland.(Xie et al., 2023)

**Point 3:**

The current dataset mainly covers the China region. It is suggested that the authors discuss whether the dataset applies to other countries with significant differences in climate or cropping patterns (e.g., Africa, Europe, and the United States). In addition, the authors need to consider whether the global data can be expanded in the future.

**Response 3:**

Thank you for your comment. Regarding your question, I will answer it from the following two aspects:

(1) From the perspective of the generalizability of the dataset construction process, we believe the process has strong general applicability. China's vast territory spans multiple climate zones, and its geographic and climatic diversity nearly encompasses the main terrain and climate features of other countries. We believe that the descriptive keywords we have designed for farmland can comprehensively cover various cropland morphologies. Although there are differences in climate and cropping practices between China and some other countries, our textual annotation framework is highly flexible. We can adjust the keywords based on the cropland climate of different regions, allowing it to adapt to the farmland characteristics of various regions around the world.

(2) From the perspective of the generalizability of the model trained on FarmSeg-VL, we have further supplemented the relevant experiments in Appendix E of the revised manuscript. The results indicate that, despite FarmSeg-VL being constructed based on remote sensing imagery from China, it still exhibits strong transferability under different climate and cropping pattern conditions, which partially validates the dataset's applicability in broader agricultural scenarios.

Furthermore, we fully agree with the reviewer's suggestion regarding global expansion. In future work, we plan to further collect and organize remote sensing imagery and farmland annotation information from various countries and regions, continually expanding the spatial coverage of the dataset to support more diverse and globally adaptable agricultural intelligence analysis research. The detailed modifications are as follows:

**Appendix**

**E Cross-Regional Applicability Assessment of FarmSeg-VL.**

To verify the generalization performance of the model trained using FarmSeg-VL on datasets from other countries that have significant differences in climate or cropping patterns compared to FarmSeg-VL, this paper selects a portion of the region in Nordrhein-Westfalen, Germany as the benchmark for testing, test experiments were conducted using the LISA model. Specifically, we selected a subset of data from Nordrhein-Westfalen, Germany, and performed several preprocessing steps, including image downloading, vector boundary processing, and image and label cropping, to adapt it for our farmland segmentation model, the image and label overlay results of the test area are shown in Fig.28.

[Figure]

**Fig. 28. Example of Fiboa data.**

The experimental results are shown in Table 8, where we compare the cross-domain performance of the LISA model trained on the FarmSeg-VL dataset with that of other models evaluated on public datasets in Section 4.4. Specifically, the FGFD and LoveDA datasets are from China, while the DGLC dataset covers regions in Thailand, Indonesia, and India. As shown in the table, the LISA model performs well in cross-domain testing, which can be attributed to the extensive geographical coverage and rich seasonal variations of the FarmSeg-VL dataset, providing a solid foundation for cross-domain feature learning. Notably, the LISA model outperforms other models on the Fiboa dataset. This is due to the concentrated, contiguous, and well-defined characteristics of farmland in the Fiboa region, which facilitate the extraction of discriminative features, leading to optimal results in this region. Furthermore, the climatic and cropping system differences between the Fiboa dataset and FarmSeg-VL further validate the applicability and strong generalization capability of the FarmSeg-VL dataset in adapting to the diverse agricultural contexts of different countries. This highlights its potential in global, heterogeneous farmland scenarios.

**Table 8. Farmland segmentation results of different methods on fiboa.**

| Evaluation Metrics(%) | LISA | | | |
| --- | --- | --- | --- | --- |
| | FGFD | LoveDA | DGLC | Fiboa |
| mACC | 83.33 | 81.76 | 72.23 | **88.05** |
| mIoU | 70.58 | 65.74 | 56.36 | **78.20** |
| mDice | 82.65 | 78.82 | 72.06 | **87.73** |
| Recall | 83.87 | 80.75 | 72.44 | **87.38** |

**Point 4:**

The text mentions that the data cover four seasons, but it is not clear whether the full growth cycle of different crops is covered. It is suggested that additional clarification be provided.

**Response 4:**

Thank you for your comment. In the process of data selection, this study has thoroughly considered the seasonal characteristics of farmland and the key stages of the crop growth cycle. Although the data encompasses all four seasons—spring, summer, autumn, and winter—it does not cover the complete growth cycle of all crops. Instead, we selectively focused on key periods when remote sensing imagery exhibited typical texture patterns for different crops in various regions. For example, in the Northeast Plain agricultural region, summer is the peak growth period for major crops, with distinct farmland texture features. Therefore, we primarily collected summer imagery data from the Northeast Plain region, including Heilongjiang and Jilin provinces, to better capture the farmland characteristics during this typical period. By selecting representative temporal imagery, we can effectively enhance the model's ability to recognize farmland spatial distribution and seasonal changes.

**Point 5:**

It is suggested to add the performance comparison of the model on the training set and the test set, or analyze the influence of data distribution on the robustness of the model through cross-validation.

**Response 5:**

Thank you for your comment. The training set, test set, and validation set in the FarmSeg-VL dataset were first mixed together, and then randomly divided into three new training, test, and validation sets at a ratio of 7:2:1. The experiments were conducted based on the LISA model. To avoid the influence of random factors, each experiment was repeated three times. The results showed that the model could maintain stable performance under different data partitioning methods, indicating that the model trained on FarmSeg-VL demonstrates strong robustness and maintains high generalization capability when faced with different data distributions. The detailed modifications are as follows:

**Appendix**

**F The influence of data distribution on the robustness of the model**

To evaluate the robustness of the model under different data partitioning conditions, we conducted additional experiments using the LISA model on the FarmSeg-VL dataset. Specifically, we first merged the original training, validation, and test sets, then randomly split the combined dataset into three new training, validation, and test sets following a 7:2:1 ratio. This random splitting procedure was repeated three times to minimize the impact of stochastic variation, and the model was trained and evaluated independently for each split.

Table 9 shows the results of four different random partitions of the test set. Test1−Test4 represent the results of four different test sets. As shown in the Table 9, the variation in test results across the different test sets is minimal, demonstrating the robustness of the FarmSeg-VL dataset and the model's robustness. This outcome indicates that the balanced distribution and diverse geographical features of the dataset play a crucial role in enhancing the model's stability and generalization capability. Specifically, the FarmSeg-VL dataset is characterized by high-quality image and textual annotations, with a broad distribution that spans different seasons and geographical conditions. This effectively reduces the discrepancies between the datasets, thereby improving the model's robustness to variations in data partitions.

**Table 9. Farmland segmentation results on different tests.**

| Evaluation Metrics(%) | Test1 | Test2 | Test3 | Test4 |
|---|---|---|---|---|
| mACC | 87.71 | 87.27 | 87.33 | 87.54 |
| mIoU | 93.47 | 93.22 | 93.26 | 93.37 |
| mDice | 93.45 | 93.20 | 93.23 | 93.36 |
| Recall | 93.46 | 93.20 | 93.24 | 93.34 |

**Point 6:**

Tables 4~11 and Figures 9~16 clearly show the situation of each agricultural region, but they occupy more space, I suggest the authors put this part in the supplementary materials.

**Response 6:**

Thank you for your comment. We have moved Tables 4–11 and Figures 9–16 from the main text to Appendix C of the manuscript, retaining essential descriptions and analyses in the main body to ensure that readers can access the necessary information without compromising the readability of the paper.

**Point 7:**

Considering the wide international readership, I suggest the authors add some non-Chinese references.

**Response 7:**

Thank you for your comment. We understand the importance of diverse international references. We would like to clarify that, although some of the cited references were published in Chinese journals, all references are written in English, and no Chinese-language sources are cited in the manuscript. In future research, we will continue to

focus on expanding our international perspective to better serve a global audience.

**Reviewer 2:**

Understanding the spatiotemporal characteristics of farmland is essential for accurate farmland segmentation. This study introduced language-based descriptions of farmland and developed FarmSeg-VL dataset, which was the first fine-grained image-text dataset for farmland segmentation. The proposed method is innovative and dataset is of high accuracy, which had great potentials as a standard benchmark for farmland segmentation. However, there are still some problems that deserve to solve before publications.

**Response:** Thank you for your insightful and encouraging comments. We appreciate your recognition of the importance of understanding the spatiotemporal characteristics of farmland, and your acknowledgment of the innovation and potential impact of the FarmSeg-VL dataset as a fine-grained image-text benchmark for farmland segmentation. We are also grateful for your recognition of the accuracy and applicability of our proposed method. In response to your suggestions, we have carefully revised the manuscript to address the remaining issues prior to publication. Specifically, we have refined methodological explanations, clarified the scope of application, and improved the consistency and clarity of language expression throughout the paper. We hope these revisions sufficiently address your concerns and further strengthen the contribution of our work.

**Point 1:**

In the introduction part, the author mentioned that the label-driven paradigm had some disadvantages for farmland segmentation. The vision-language models (VLMs) can capture more contextual and background information from imageries. More information about the construction of VLM model can be included.

**Response 1:**

Thank you for your valuable comment. We have revised the introduction to include a clearer and more detailed explanation of the construction of VLM model. The detailed modifications are as follows:
* * *
**1 Introduction**

With the emergence of vision language models (VLMs) and their expanding applications across various fields, studies (Devlin et al., 2019; Wu et al., 2025b, a) have shown that language can reveal deeper semantic clues behind visual information. These VLMs typically follow a general construction process: first, feature representations are extracted from images through a visual encoder, a process aimed at capturing key visual representation in the images. For example, in the LLaVA model (Liu et al., 2023), the image representations generated by the fixed visual encoder lay the foundation for subsequent processing. Next, to establish a connection between vision and language, the model needs to map the extracted visual features to the space of the language model, enabling visual representation

to be translated into natural language descriptions or understood. The LLaVA model precisely utilizes this method, mapping image representations to the prompt space of large language models, helping the model understand the relationship between visual representation and linguistic expressions, thereby achieving efficient downstream tasks. Furthermore, to enable the model to handle complex tasks, integrating visual perception with language understanding becomes a key step. LISA (Lai et al., 2023) is a typical example, it not only combines visual perception capabilities but also incorporates in-depth language understanding abilities, allowing it to perform reasoning-based tasks such as segmentation tasks. This multimodal information processing capability is one of the important characteristics of VLMs, enabling them to consider visual context while understanding and generating language. These breakthroughs make up for the shortcomings of relying solely on label-guided models to handle complex spatiotemporal heterogeneous farmland scenes, making it possible to mine the complex semantic information in farmland remote sensing images and then model the deep inherent logical relationship between farmland and its surroundings.

**Point 2:**
The image-text datasets is also the core of the VLM model, and what is the difference with the traditional label-driven deep learning method.

**Response 2:**
  Thank you for your valuable comments. The differences between traditional label-driven deep learning methods and the VLM-based approach are as follows:
(1) The existing label-driven paradigm primarily focuses on learning the shape and texture features of the farmland and its surrounding environment, neglecting the inherent logical relationships between farmland and its environment in complex agricultural scenes. As a result, when the model is tasked with segmenting farmland in complex scenarios, it fails to capture the deep connections between farmland and its environment, leading to insufficient generalization ability. In contrast, language can describe environmental features such as buildings, water resources, and vegetation layout around the farmland. Guided by language, VLMs enhance the inherent logical connections between farmland and surrounding land features.
(2) Labels often fail to fully reflect the evolving characteristics of farmland across different seasons and growth stages, leading to insufficient generalization ability of traditional label-driven deep learning methods in spatiotemporal dynamic scenarios. In contrast, language, by describing information such as seasonal changes and spatial location of the farmland, provides the model with rich prior knowledge. As a result, VLMs are able to capture the phenological features of farmland as it changes with the seasons, climate fluctuations, and crop growth cycles, while also learning the morphological differences caused by geographical variations. This enables a better understanding of the spatiotemporal dynamic changes of farmland, effectively alleviating the segmentation challenges posed by temporal and spatial variations.

**Point 3:**
  In table 1, the abbreviation should be explained, such as SR for spatial resolution.

**Response 3:**
  Thank you for your valuable comments. Based on your suggestions, we have made

the necessary abbreviations modifications in Table 1. "SR" has been changed to "Spatial Resolution," and "FP" now represents "Farmland Proportion." The detailed modifications are as follows:

**Table 1 Detailed information on non image-text dataset of farmland.**

| Type | Dataset | Category | Spatial Resolution | Image size | Farmland Proportion | Region |
|------|---------|----------|--------------------|-----------|--------------------|--------|
| Non-dedicated datasets | Evlab-SS | 11 | 0.1-2 | 4500×4500 | 8.77 | / |
| | GID | 15 | 4 | 56×56,112×112,224×224 | 30.66 | China |
| | DGLC | 7 | 0.5 | 2448×2448 | 57.74 | Thailand, Indonesia, India |
| | LoveDA | 7 | 0.3 | 1024×1024 | 26.79 | Nanjing,Changzhou,Wuhan, China |
| | Bigearthnet | 43 | 10-60 | 120×120 | 12.41 | / |
| Dedicated datasets | GFSAD30 | 3 | 30 | / | / | Europe,Middle East,Russia and Asia |
| | VACD | 2 | 0.5 | 512×512 | / | Guangdong,China |
| | WEIMIN | 2 | 0.5-2 | 512×512 | / | Hebei,China |
| | FGFD | 2 | 0.3 | 512×512 | / | Heilongjiang,Hebei,Shanxi, Guizhou,Hubei,Jiangxi,Xizang,China |

**Point 4:**

In section 2.2, the author mentioned that the most of image-text datasets just described scene level or object level characteristics instead of specific like farmland segmentation. I wonder the inherent difference among these textural descriptions, and authors can take some detailed examples.

**Response 4:**

Thank you for your valuable comments. To further illustrate the limitations of farmland texture descriptions in existing image-text datasets and the differences with our dataset, we have added several specific examples and analyzed the inherent differences in terms of granularity and task suitability. The relevant supplements have been placed in the appendix, the detailed modifications are as follows:

**Appendix**

**A More details of farmland texture description in image-text dataset**

As shown in Fig.9, mainstream remote sensing image-text datasets, such as UCM-Captions, NWPU-Captions, RSICD, RSICap, and ChatEarthNet, generally adopt scene-level or object-level descriptions. These datasets often lack detailed characterization of farmland morphology, temporal features, and environmental context, making them insufficient for farmland segmentation tasks that require high-level semantic and structurally rich textual information.

For example, UCM-Captions provides only simple and repetitive descriptions like "There is a piece of farmland," without any specific texture or spatial information. NWPU-Captions offers slight improvements by adding color and shape descriptions, such as "Many dark green circular fields are mixed with yellow rectangular fields," but still does not include background context or agricultural semantics. RSICD focuses only on aggregated forms or land cover components, with descriptions like "The little farm is made up of grass and crops," lacking both temporal cues and environmental context. RSICap provides relatively richer descriptions, for example, "In the image, there are many buildings and some farmlands located near a river," which reflects spatial relationships between farmland and buildings or water bodies. However, these descriptions are mostly static and fail to capture the dynamic properties of farmland over time. ChatEarthNet, designed primarily for land cover classification, presents slightly more complex descriptions such as "This image shows a balance between crop and grass areas.", but still lacks detailed information about farmland morphology, terrain, crop types, or surrounding environmental elements. In contrast, the proposed FarmSeg-VL dataset is specifically designed for the farmland segmentation task, placing greater emphasis on fine-grained semantic information closely tied to the spatiotemporal characteristics of farmland. For each remote sensing image, the accompanying textual description includes the image capture time, geographic coordinates, and detailed references to landform, shape, boundary characteristics, topography, as well as surrounding features such as water bodies, vegetation, and buildings. Additionally, the descriptions incorporate attributes such as cropping patterns and spatial layouts, providing comprehensive semantic support for accurate and context-aware farmland segmentation.

[Figure]

**Fig. 9. Details of farmland texture description in general remote sensing image-text dataset.**

**Point 5:**

China has a vast territory and different regions have different agricultural conditions. The construction of FarmSeg-VL dataset and related textural descriptions should consider this.

**Response 5:**

Thank you for your comment. We have considered the agricultural diversity across different regions of China in the construction of the FarmSeg-VL dataset and related textual descriptions. Specifically, this is addressed in Section 3.1, under 1) RS Image Acquisition and Processing and 2) Caption Construction. We apologize for the unclear description, and we have made appropriate revisions to enhance readability for the readers. The specific changes are as follows:
* * *
**1) RS Image Acquisition and Processing**

China's vast territory, diverse landforms, and complex climate result in significant regional variations in agricultural conditions, leading to highly heterogeneous texture features and distribution patterns of farmland in remote sensing imagery. As a result, farmland exhibits significant spatiotemporal dynamics and fragmented distribution characteristics, presenting diverse spatial patterns due to these regional differences. For example, the land in the Northeast China Plain is flat and fertile, and the farmland has the characteristics of concentrated distribution and regular shape, while the Yungui Plateau in China has complex terrain and diverse climate, and the farmland has the characteristics of dispersed distribution and fragmented shape. The farmland appearance and characteristics of these agricultural areas are unique, which poses different challenges and opportunities for farmland segmentation. This study selected representative agricultural regions based on the spatial distribution and morphological characteristics of farmland. Specifically, based on the spatial aggregation and morphological regularity of farmland, the Northeast China Plain and Huang-Huai-Hai Plain were selected as typical regions characterized by concentrated and regular-shaped farmland. For areas with sloped farmland distribution, the Northern Arid and Semi-Arid Region and the Loess Plateau were chosen as study areas. At the same time, in view of the particularity of farmland morphology, such as narrow and long, striped, and sporadic and fragmented, the South China Areas, Sichuan Basin, Yungui Plateau, and Yangtze River Middle and Lower Reaches Plain were selected as research areas. The study covers 13 provincial-level administrative regions, including Heilongjiang, Jilin, Ningxia, Hebei, Henan, Shandong, Shaanxi, Anhui, Hunan, Jiangsu, Guangdong, Sichuan, and Yunnan. These regions provide broad spatial coverage, highlight distinct regional characteristics, and are highly representative and typical of China's diverse agricultural landscapes.

**2) Caption Construction**

For the caption construction of each farmland sample, this study summarizes 11 key elements for describing farmland: shape, boundary morphology, shooting time, sowing conditions, the macro-level distribution of farmland, geographic location information, topographical features, landscape, the distribution of buildings, water bodies, and vegetation. The spatiotemporal characteristics of farmland result from the interaction of multiple factors.(Wang et al., 2022b) Temporally, the variations in crop growth stages lead to distinct visual texture differences in farmland across different seasons. (Zhu et al., 2022) Spatially, farmland exhibits significant spatial differentiation, with different regions affected by factors such as topography, terrain, and water-heat conditions, resulting in noticeable variations in farmland morphology and layout. (Pan and Zhang, 2022) Therefore, this study considers the issue at multiple spatial scales. At the macro-regional scale, typical farmland images

were collected from various agricultural regions across China. These regions are not only located in different latitudes and longitudes, but also have different terrains and topography. For instance, farmland in the Northeast China Plain is flat and typically follows a concentrated distribution pattern with regular shapes, which is reflected in descriptions such as "the farmland primarily exhibits concentrated contiguous distribution" and "the shape of the farmland is characterized by blocky." In contrast, the terrain of South China is predominantly hilly and mountainous, leading to a more dispersed farmland distribution and irregular shapes, which is described in the text as "with the farmland primarily in a dispersed distribution" and "the terrain is undulating." Similarly, farmland in regions like the Loess Plateau and the arid and Semi-Arid Northern Areas often displays terraced or sloping patterns. At the same time, the spatial coupling relationships between farmland and surrounding features, such as water bodies and buildings, are key factors influencing the distribution and accuracy of farmland identification.(Duan et al., 2022; Zheng et al., 2022) The relationship between the farmland and surrounding environmental features is expressed, for example, as "the water bodies surrounding the farmland mainly consist of scattered blocky ponds," and "the vegetation around the farmland mainly consists of scattered trees and scattered forests. " Similarly, the segmentation of farmland relies on boundary and texture information, the shape of the farmland and the boundary morphology, is also crucial for accurate identification of farmland. (Xie et al., 2023)

**Point 6:**

In figure 2, the authors should an example of 5 types of text characteristics for farmland fragmentation. In future study, more quantitative description can be included.

**Response 6:**

Thank you for your valuable comments. In response to your request to add five examples of farmland fragmentation text features in Figure 2, we have included the corresponding examples in the appendix to further assist readers in understanding farmland shape characteristics. In future studies, from the perspective of the textual descriptions of farmland samples, we may add proportional descriptions, such as farmland area ratio and fragmentation degree, to provide a more comprehensive assessment of farmland. These additional metrics could include the mean patch size of farmland, the proportion of farmland area to total image area, and the number of fragmented patches per hectare, which will further deepen the understanding of farmland fragmentation and its spatial characteristics.   From the perspective of the dataset's scale, and coverage, FarmSeg-VL covers 13 provinces in China, spanning approximately 4,300 km². We plan to add data from 2 to 3 other countries (e.g., Africa, Europe, and the United States) to expand our dataset, aiming for global diversification. The detailed modifications are as follows:

**Appendix**

**B Examples of 5 types of text features for farmland shapes**

To provide readers with a more intuitive understanding of the farmland morphology in the FarmSeg-VL dataset, we present five additional examples of farmland shapes in Fig. 18, beyond those shown in Fig. 10.

[Figure]

**Fig. 10. Example of farmland shape.**

**Point 7:**

Table 4-11 can be converted into the figure to improve the readability. What is the LoveDA dataset in section 4.5?

**Response 7:**

We have adopted your suggestion and converted Tables 4-11 into visualized charts (see Fig. 11 to 18 in the revised manuscript), significantly enhancing the intuitiveness and readability of the results. The LoveDA dataset is a publicly available land cover classification dataset that was presented at the NeurIPS 2021 Datasets and Benchmarks Track. Since this paper focuses on farmland segmentation tasks, in the experiment of Section 4.5, we adapted the dataset by merging its original multiple non-farmland categories into a unified background label, retaining only the "farmland" and "non-farmland" categories. This modification aligns its label system with our binary segmentation task. The detailed modifications are as follows:

**Appendix**

**C Farmland segmentation results of different methods in different agricultural areas**

**Northeast China Plain**

[Figure]

**Fig. 11. Farmland segmentation results of different methods in Northeast China Plain.**

**Huang-Huai-Hai Plain**

[Figure]

**Fig. 12. Farmland segmentation results of different methods in Huang-Huai-Hai Plain.**

**Northern Arid and Semi-arid Region**

[Figure]

**Fig. 13. Farmland segmentation results of different methods in Northern Arid and Semi-arid Region.**

**Loess Plateau**

[Figure]

**Fig. 14. Farmland segmentation results of different methods in Loess Plateau.**

**Yangtze River Middle and Lower Reaches Plain**

[Figure]

**Fig. 15. Farmland segmentation results of different methods in Yangtze River Middle and Lower Reaches Plain.**

**South China Areas**

[Figure]

**Fig. 16. Farmland segmentation results of different methods in South China Areas.**

**Sichuan Basin**

[Figure]

**Fig. 17. Farmland segmentation results of different methods in Sichuan Basin.**

**Yungui Plateau**

[Figure]

**Fig. 18. Farmland segmentation results of different methods in Yungui Plateau.**

**Point 8:**

In the abstract and conclusion parts, the authors should add more information and quantitative results about the farmland segmentation results in this study.

**Response 8:**

Thank you for your valuable comments. In the abstract and discussion, we have added the quantified results of farmland segmentation. The detailed modifications are as follows:

**Abstract.** Understanding and mastering the spatiotemporal characteristics of farmland is essential for accurate farmland segmentation. The traditional deep learning paradigm that solely relies on labeled data has limitations in representing the spatial relationships between farmland elements and the surrounding environment. It struggles to effectively model the dynamic temporal evolution and spatial heterogeneity of farmland. Language, as a structured knowledge carrier, can explicitly express the spatiotemporal characteristics of farmland, such as its shape, distribution, and surrounding environmental information. Therefore, a language-driven learning paradigm can effectively alleviate the challenges posed by the spatiotemporal heterogeneity of farmland. However, in the field of remote sensing imagery of farmland, there is currently no comprehensive benchmark dataset to support this research direction. To fill this gap, we introduced language-based descriptions of farmland and developed FarmSeg-VL dataset—the first fine-grained image-text dataset designed for spatiotemporal farmland segmentation. Firstly, this article proposed a semi-automatic annotation method that can accurately assign caption to each image, ensuring high data quality and semantic richness while improving the efficiency of dataset construction. Secondly, the FarmSeg-VL exhibits significant spatiotemporal characteristics. In terms of the temporal dimension, it covers all four seasons. In terms of the spatial dimension, it covers eight typical agricultural regions across China, with a total area of approximately 4,300 km². In addition, in terms of captions, FarmSeg-VL covers rich spatiotemporal characteristics of farmland, including its inherent properties, phenological characteristics, spatial distribution, topographic and geomorphic features, and the distribution of surrounding environments. Finally, we perform a performance analysis of the vision language model and a deep learning model that relies only on labels trained on FarmSeg-VL. Models trained on the vision language model outperform deep learning models that rely only on labels by 10%-20%, demonstrating its potential as a standard benchmark for farmland segmentation. The FarmSeg-VL dataset will be publicly released at https://doi.org/10.5281/zenodo.15099885(Tao et al., 2025).

**6 Conclusion**

This study constructs FarmSeg-VL, a high-quality image-text dataset specifically designed for farmland segmentation, with key features including high-precision images and masks, extensive spatiotemporal coverage, and refined captions of farmland characteristics. In the dataset construction process, Google imagery with a resolution of 0.5-2 meters was selected as the image data source. Through in-depth analysis of numerous farmland samples, five key attributes were summarized: inherent properties, phenological characteristics, spatial distribution, topographic and geomorphic features and distribution of surrounding environments. These were further refined into 11 specific descriptive dimensions, covering shape, boundary patterns, season, sowing situation, geographic location, distribution, terrain, landscape features, as well as the distribution of water bodies, buildings, and trees in the surrounding environment. Based on the above keywords, a farmland description template was designed, and a semi-automated annotation method was used to generate binary mask labels and their corresponding captions for each image. Ultimately, a dedicated dataset consisting of

22,605 image-text pairs was constructed. To verify the advantages of the FarmSeg-VL in enhancing farmland segmentation accuracy compared to general image-text datasets, this study first conducted fine-tuning experiments on three leading vision language segmentation models: LISA, PixelLM, and LaSagna. The experimental results demonstrate that the model fine-tuned with the FarmSeg-VL significantly outperforms the model trained with general image-text datasets in segmentation performance. Additionally, this study compared the VLMs trained on the FarmSeg-VL to a traditional deep learning model that relies solely on labels. The results show a 10% to 20% improvement in segmentation accuracy across different agricultural regions and datasets, highlighting that language guidance effectively mitigates the impact of spatiotemporal heterogeneity on farmland segmentation. Finally, the study compared the performance of the traditional deep learning model relies solely on labels trained on the FGFD dataset with the models trained using three VLMs on the FarmSeg-VL dataset. The evaluation on the LoveDA dataset showed an improvement in test accuracy by approximately 15%. Experimental results show that the model trained on FarmSeg-VL significantly improves accuracy and robustness in farmland segmentation. As the first large-scale image-text dataset for farmland segmentation, FarmSeg-VL holds significant academic value and application potential. It is expected to advance research on semantic understanding of farmland in remote sensing imagery, promote the development of more efficient and generalized segmentation models, and better serve the diverse needs of agricultural monitoring.

**Point 9:**

I wonder if the author adds some negative samples for the farmland fragmentation? Table show the comparison results solely on labels and VLMs. The deep learning models for these two kinds are different and the training samples for these two kinds are also different. How could these be compared?

**Response 9:**

Thank you for your valuable comments. This study focuses on farmland binary classification, where the image and mask data only include annotations for cropland and non-farmland areas. The non-farmland class, which is labeled as the background, represents the negative samples. In the textual descriptions, we have incorporated rich semantic information about the surrounding environment, which serves as the description for the negative samples. These descriptions include keywords such as "blocky ponds," "meadow," and "scattered buildings," which help the model better understand the relationship between cropland and surrounding land features. Regarding the rationale for model comparison, we understand the reviewers' concerns about comparing different architectures and training sample variations. The core purpose of this comparison is to assess the impact of introducing textual description information on farmland segmentation performance. In the training samples, the images and masks of farmland remain consistent, and only the text description of farmland is added to VLM. Although the model architectures and training data details differ, the comparison is based on one key point: the VLM only adds natural language text during training in addition to the label-based deep learning method, while the visual input and visual encoding remain consistent. During testing, the prompt for the VLM was "Please segment the farmland in the image," which aligns with the goal of the label-based method that relies solely on visual information for inference.

**Point 10:**

There are still some grammatical and lingual problems, and authors should make a thorough revision, such as "annotationand" in line 115.

**Response 10:**

Thank you for your valuable comments. We have carefully reviewed the entire manuscript and made comprehensive revisions to address the grammatical and language issues, including correcting "annotationand" to "annotation and" in line 115.